# Co-coding of head and whisker movements by both VPM and POm thalamic neurons

Tess Baker Oram [1,2], Alon Tenzer[1,2], Inbar Saraf-Sinik[1], Ofer Yizhar [1] &
Ehud Ahissar [1] ✉

Rodents continuously move their heads and whiskers in a coordinated manner while perceiving objects through whisker-touch. Studies in head-fixed rodents showed that the ventroposterior medial (VPM) and posterior medial (POm) thalamic nuclei code for whisker kinematics, with POm involvement reduced in awake animals. To examine VPM and POm involvement in coding head and whisker kinematics in awake, head-free conditions, we recorded thalamic neuronal activity and tracked head and whisker movements in male mice exploring an open arena. Using optogenetic tagging, we found that in freely moving mice, both nuclei equally coded whisker kinematics and robustly coded head kinematics. The fraction of neurons coding head kinematics increased after whisker trimming, ruling out whisker-mediated coding. Optogenetic activation of thalamic neurons evoked overt kinematic changes and increased the fraction of neurons leading changes in head kinematics. Our data suggest that VPM and POm integrate head and whisker information and can influence head kinematics during tactile perception.

Sensory vibrissal signals are conveyed in the brain primarily via the lemniscal and paralemniscal pathways, which afferent via the VPM and POm thalamic nuclei, respectively[1]. Previous studies in anesthetized and awake head-fixed rodents[2–7] have resulted in conflicting conclusions regarding the involvement of the POm, and the paralemniscal pathway, in the processing of active vibrissal information. In anesthetized head-fixed rodents, VPM neurons were found to code both whisking (self-motion) and touch (contact with external objects) information, whereas POm neurons were found to code only whisking information[4,8]. In contrast, in awake head-fixed rodents, POm neurons were found to be almost non-responsive to whisking signals[6,7]. Head fixation affects whisking strategies[9] and thus might also affect thalamic processing. Additionally, due to head fixation, previous studies could not assess the involvement of VPM or POm neurons in processing head kinematics, which directly affect vibrissal signaling and tactile perception.

This study was designed to address two key questions: (1) Are POm neurons involved in the processing of vibrissal signals during head-free whisking? (2) How do head kinematics affect thalamic processing in both VPM and POm?

The VPM and POm share an anatomical border that cannot be accurately defined during or after neuronal recordings[4]. Moreover, extracellular signals from a neuron on one side of the border can often be recorded through an electrode tip residing more than 50 microns into the other side. In addition, not all neurons belonging to the lemniscal or paralemniscal systems necessarily adhere to these anatomical borders. In contrast, the molecular signatures of these two systems differ in several clear aspects[10].

In this work, we applied optogenetic tagging to distinguish between VPM and POm neurons in freely exploring mice. In these mice, POm and VPM neurons were similarly involved in whisker kinematic coding, a pattern that is more consistent with the anesthetized head-fixed condition than with the awake head-fixed condition. Additionally, neurons in both thalamic nuclei robustly coded head kinematics – azimuth, azimuthal velocity, and linear speed. Given that these neurons can potentially affect motor functions via several anatomical pathways[11–17], we tested the effect of their stimulation on head kinematics and found that these thalamic neurons can effectively participate in a perceptual closed-loop that includes head kinematics.

---

[1]Department of Brain Sciences, Weizmann Institute of Science, Rehovot, Israel. [2]These authors contributed equally: Tess Baker Oram, Alon Tenzer.
✉e-mail: ehud.ahissar@weizmann.ac.il

## Results

We applied optogenetic tagging to distinguish between neuronal units affiliated with the lemniscal and paralemniscal pathways. To selectively target the expression of opsins and fluorescent proteins to either VPM or POm neurons, we used the GPR26 transgenic mouse line, which strongly expresses Cre in the POm but not the VPm[18] (Supplementary Fig. 1). In these mice, a titer-matched mixture of a viral construct with Cre-ON expression of a red fluorescent protein (AAV8. EF1α. DIO.hM4D-mCherry) and Cre-OFF expression of yellow fluorescent protein (AAV2/1.hSyn. DFO.ChR2-EYFP; Fig. 1a) injected into the POm and VPM thalamic nuclei led to the expression of mCherry in the POm, and EYFP in the VPM, and similarly in their downstream cortical targets (Fig. 1b and Supplementary Fig. 2). Channelrhodopsin expression in the POm (using a Cre-On construct AAV2/1.EF1α.DIO.ChR2-EYFP) or in the VPM (using the Cre-Off construct AAV2/1.hSyn. DFO.ChR2-EYFP) and a chronic implantation of a multi-site optrode allowed us to stimulate and record the activity of thalamic neurons in "POm mice" and "VPM mice", respectively (Supplementary Fig. 3)[19].

Responses to laser pulses (Fig. 1c) exhibited a bimodal distribution of onset latencies (Fig. 1d) in these mice. Based on these data, neurons were classified as "tagged" if their response intensity crossed a pre-defined threshold and their onset latency was <6 ms (see "Methods" section), indicating that photostimulation likely excited the neuron itself rather than nearby connected neurons[20]. We further used our setting to address a long-standing question: do differences in adaptation patterns between POm and VPM[3] stem from afferent or intrinsic thalamic differences? We thus tested the adaptation of VPM and POm neurons in response to repetitive whisker stimulation or direct optogenetic excitation when the mice were anesthetized. Interestingly, whereas POm neurons exhibited stronger adaptation than VPM neurons in response to relatively rapid (≥5 Hz) whisker stimulation, as previously observed[3], both populations showed similar adaptation to direct optogenetic excitation (Fig. 1e, single cells and

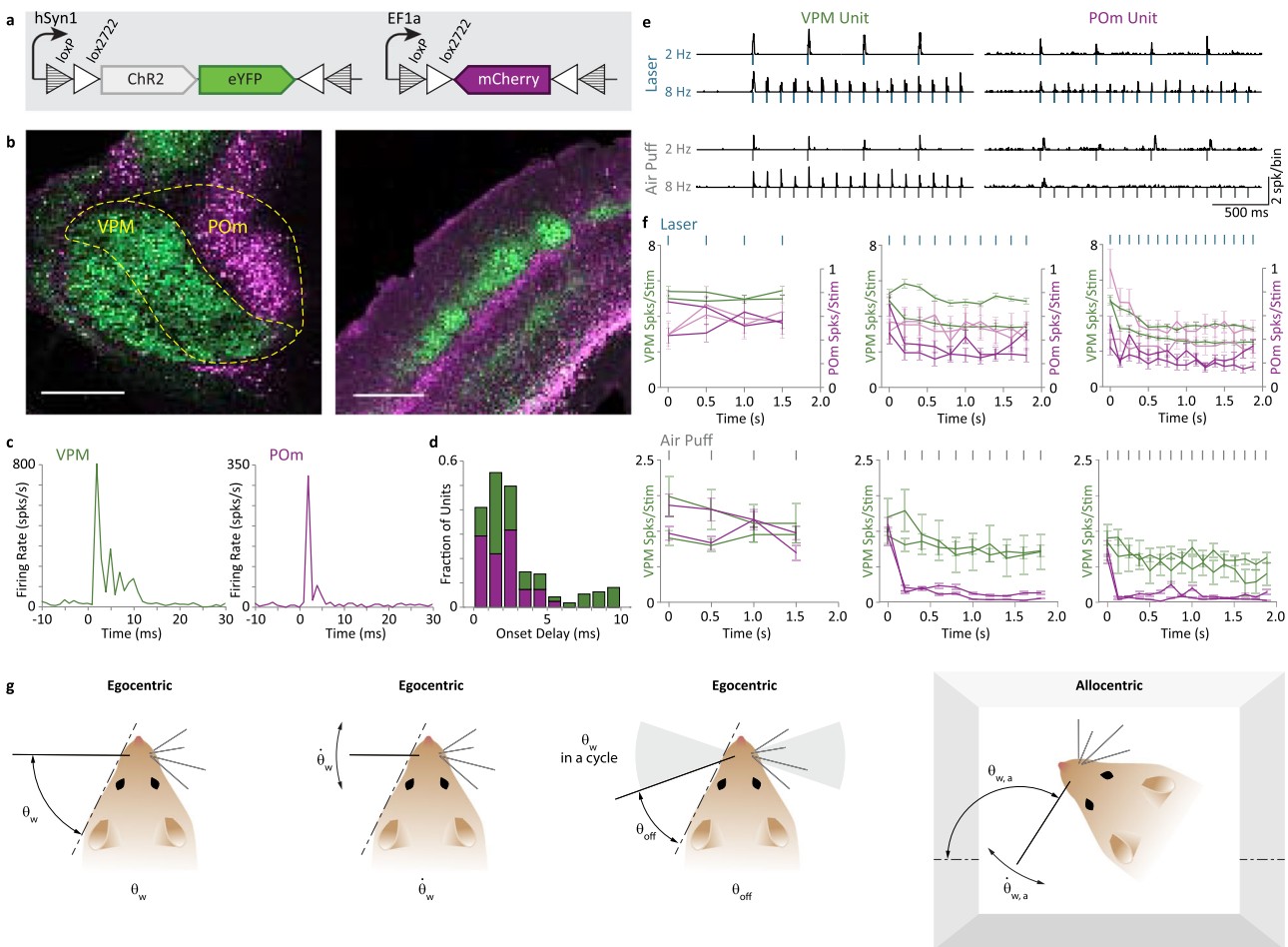

**Fig. 1 | Optogenetically tagged recording from POm and VPM. a** Optogenetic tagging. Top, schematics of the Cre-OFF (left) and Cre-ON (right) viral constructs used to test the efficacy of the Cre-ON/Cre-OFF genetic targeting strategy. **b** Left, the Cre-ON/Cre-OFF genetic targeting strategy was verified by simultaneously and differentially targeting the expression of EYFP to the VPM (green) and mCherry to the POm (purple). Right, pathway-specific thalamocortical projections of EYFP-expressing VPM neurons (green) and mCherry-expressing POm neurons (purple) can be seen throughout the barrel cortex. Scale bars: 0.5 mm. This experiment was repeated 4 times with similar results. **c** PSTHs of responses (mean of 48 repetitions) of a VPM neuron (left, green) and a POm neuron (right, purple) to laser stimulations at 2 Hz (10 ms pulses) under anesthesia (see "Methods" section). **d** Distribution of onset latencies. Latencies (to 10% of response peak) ranged between 0.5 and 24 ms, with 96% of them ranging between 0.5 – 10 ms; their fraction within each nucleus

are depicted (stacked) for the range 0–10 ms. Neurons whose onset latency belonged to the first mode of the distribution ( < 6 ms) were classified as directly-tagged. **e** Left, response of a directly-tagged VPM neuron to 2 Hz and 8 Hz 460 nm blue laser stimulation (blue ticks) and to 2 Hz and 8 Hz air puff stimulation of the whiskers (gray ticks). Right, response of a directly-tagged POm neuron to the same stimuli. Bin size is 2 ms. **f** Responses of directly-tagged VPM (green traces, *n* = 66 units) and POm (purple traces, *n* = 23 units) evoked by 10 ms pulses of 460 nm blue laser stimulation (blue ticks) and whisker air puff stimulation (gray ticks; VPM *n* = 10 units; POm *n* = 45 units). Laser intensities: VPM, 30 mW/mm²; POm, 150 mW/mm² (light purple) and 600 mW/mm² (dark purple). Evoked spikes were counted in the 100 ms following stimulus onset. Error bars represent standard errors of the mean. **g** Illustrations of the egocentric and allocentric angular variables of whisking (see text). Source data are provided as a Source Data file.

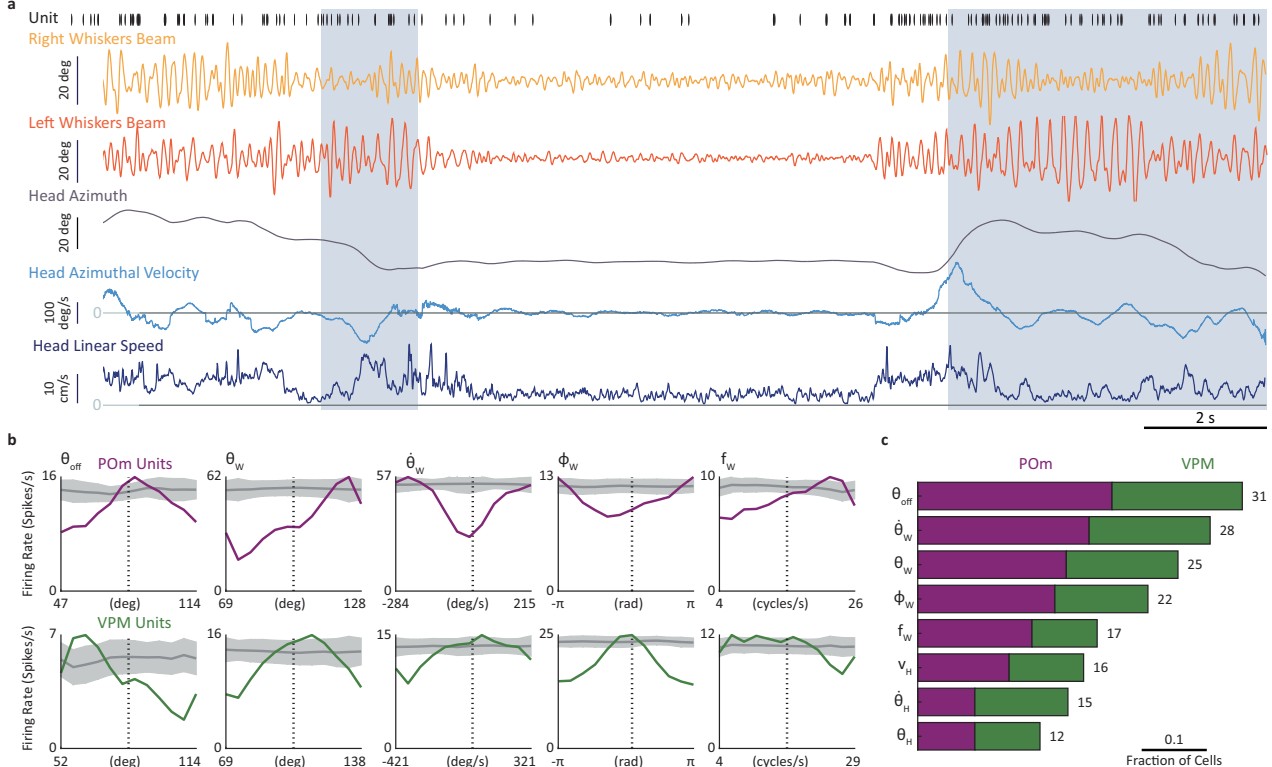

**Fig. 2 | Thalamic coding of whisking and head kinematics. a** Simultaneous recordings of head and whisker kinematics (lower traces; "whisker beams" refer to the median rotation angle of all tracked whiskers in a given side, see "Methods" section) and single-unit spike activity (top trace, each bar indicates spike time of a single unit). The gray shading indicates periods of object contact. **b** Examples of single-unit tuning curves for primary whisking-related variables (offset ($\theta_{off}$), angle ($\theta_W$), angular velocity ($\dot{\theta}_W$), phase ($\varnothing_W$) and frequency ($f_W$). Tuning curves of shuffled data are depicted by their mean (dark gray) and 1 SD (light gray). Purple, POm; Green, VPM. Each curve, of both empirical and shuffled data, is normalized to its maximal value; the firing rates labeling the ordinate in each plot refer to the empirical data only. **c** Fraction of tagged units coding each of the whisker kinematic variables (for either side of the snout) and head kinematic variables, computed for each nucleus separately (fraction values are not additive across nuclei). Number of units (n) appear on the right y-axis. Source data are provided as a Source Data file.

Fig. 1f, mean responses of neuronal populations). The similar profiles of adaptation to direct stimulation suggest that the difference in the adaptation pattern to whisker stimulations in the two pathways does not stem from differences in intrinsic adaptation mechanisms in VPM and POm neurons.

## Coding of whisking kinematics

We next recorded behavior and thalamic activity while performing a spontaneous emergence task, where the mouse was free to choose whether to stay in a home cage or emerge to an open arena[21] back-illuminated with infra-red light through a clear Plexiglass floor. Whisker and head kinematics (Fig. 1g) were tracked in the arena at 500 Hz simultaneously with single-unit activity (n = 254 units in total) from either the POm or VPM of the left hemisphere (Fig. 2a and Supplementary Movie 1). The results related to the emergence task presented here describe the behavior of directly tagged neurons (n = 70 single units in VPM, in 4 mice, and n = 57 single units in POm, in 6 mice; henceforth called "tagged neurons") and the kinematics of the entire beam of whiskers on each side of the snout, represented by the median angle of all the simultaneously tracked whiskers on each side[22].

For each neuron, we constructed response tuning curves to each kinematic variable (i.e., the relative spiking activity coinciding with each value of the kinematic variable) and compared them to those computed from temporally-permutated (henceforth "shuffled") data (Fig. 2b; see "Methods" section). A modulation depth [MD = (maximum firing rate -minimum firing rate)/mean firing rate] was computed for each tuning curve. A neuron was considered to be tuned to a specific variable if the MD of its tuning curve was significantly different ($p < 0.05$; bootstrap) from the population of tuning curves computed from its shuffled data (see "Methods" section. Note that this criterion does not imply a causal relationship between neuronal activity and kinematic changes[23,24]). Neurons in both VPM and POm were tuned to key variables of whisking (Fig. 1g): offset angle ($\theta_{off}$-the most retracted angle in each whisking cycle), angle ($\theta_W$), angular velocity ($\dot{\theta}_W$), phase ($\varnothing_W$) and frequency ($f_W$) (Fig. 2b). Interestingly, unlike in awake head-fixed rodents[6,7], but consistent with data from anesthetized head-fixed rats[4], the fractions of whisker kinematic coding neurons in freely moving mice were comparable across the two nuclei (Fig. 2c).

The distributions of the MDs and modal values (values for which the cells responded maximally) for the whisking variables across all neurons (Fig. 3a) revealed the following: (a) MDs of VPM and POm neurons showed comparable distributions, and (b) modal values covered large kinematic ranges, with POm neurons tending to fire more at larger whisking angles and higher whisking frequencies than VPM neurons. VPM and POm distributions of modal values were significantly different for $\theta_W$ and $f_W$ ($p < 0.03$) and not for the other variables ($p > 0.05$; permutation tests, see "Methods" section).

## Co-coding of head and whisker kinematics

In freely moving animals, head kinematics is a significant factor (Fig. 2a). We analyzed the tuning of thalamic neurons to three traditionally studied head kinematic variables – head azimuth in allocentric coordinates ($\theta_H$), head azimuthal velocity ($\dot{\theta}_H$) and head linear speed ($V_H$) – variables that reflect the compound head and body motion (see "Methods" section). This analysis revealed that both thalamic nuclei

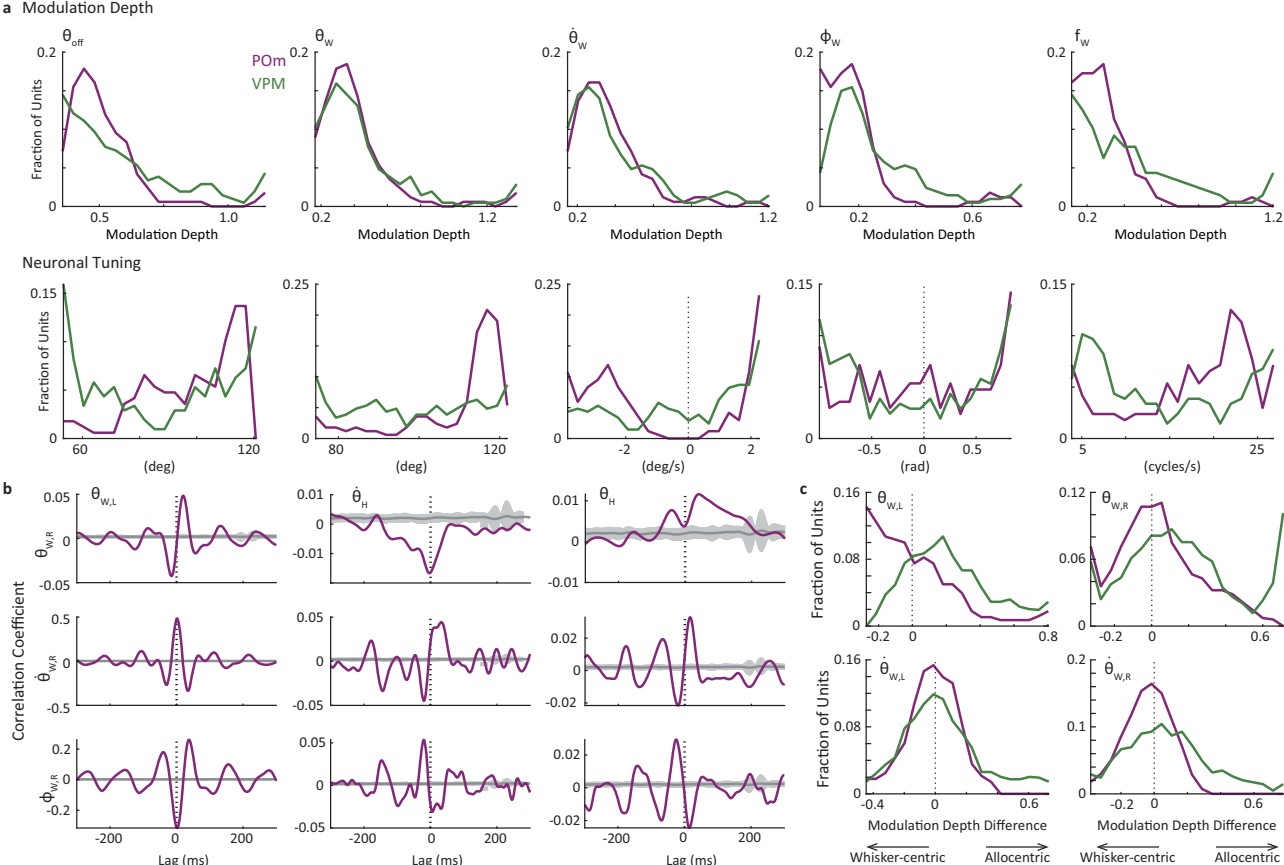

**Fig. 3 | Whisker-centric and allocentric coding in the thalamus. a** Distributions of MDs (top) and modal values (bottom) of the neuronal tunings in VPM and POm for the primary whisking-related variables. **b** Cross-correlations of head and whisker kinematic variables from the entire recording session (190.2 s) of one mouse. The triggering variable is denoted on the top-left of the top panels and the other variable along the y-axis of the left column. X-axis values are time lags in ms (resolution 2 ms); y-axis values are correlation coefficients. Correlations of shuffled data are depicted by their mean (dark gray) and 1 SD (light gray). $\theta_{W,L}$, angle of left whisker beam; $\theta_{W,R}$, angle of right whisker beam; $\dot{\theta}_{W,R}$, angular velocity of right whisker beam; $\phi_{W,R}$, phase of right whisker beam. **c** Preference of coordinate frame. The x-axis represents the MD exhibited by each neuron when coding the allocentric (whisking+head, $\theta_{W,aX}$ and $\dot{\theta}_{W,aX}$, see "Methods" section) variable minus its MD when coding the whisking variable alone (e.g., the panel labeled '$\theta_{W,L}$' represents the MD of $\theta_{W,aL}$ minus the MD of $\theta_{W,L}$). VPM and POm distributions were significantly different for $\theta_{W,L}$ ($p = 0.008$) and not for the other variables ($p > 0.08$); permutation tests. Source data are provided as a Source Data file.

carry a significant amount of information about head kinematics (Fig. 2c).

Recording times that were free of snout occlusions (caused by the mouse's headstage blocking the camera's view) and of contacts with objects in the emergence task were relatively short (total durations between 64.9 and 524.9 s per mouse; 161.9 mean ± 131.8 SD s over 10 mice). With this data set, 80/127 tagged thalamic neurons significantly coded at least one whisking or head motion variable, 68/127 at least one whisking variable, 33/127 at least one head motion variable, and 21/127 coded at least one whisking variable and at least one head motion variable. To test whether shared coding might be inherited from significant correlations between head motion and whisking variables, we analyzed the inter-variable correlations in our data set. While the correlations between the kinematic behaviors of the left and right whiskers were substantial, as expected, correlations between head and whisker kinematics were negligible in strength (magnitude of sample correlation coefficient $|r_{xy}| < 0.05$ for all pairs of head motion and whisking variables across all mice; example pairs in one mouse are shown in Fig. 3b). Thus, although head and whiskers moved in coordination, their movements were not tightly correlated at time scales relevant to tactile perceptual coding. This renders the inheritance of behavioral correlations an unlikely explanation for shared coding of head motion and whisking in the thalamus and points to neuronal integration, either in the thalamus or in networks projecting to it.

One possible function of this integration is a translation of whisker kinematics from whisker-centric to allocentric coordinates (Fig. 1g) in order to allow the coding of the actual location of objects in space[25]. We thus tested whether thalamic neurons are better tuned to either the allocentric or the whisker-centric coordinate framework. To do this, we first computed the tunings of our neurons to whisker azimuth ($\theta_{W,a}$) and azimuthal velocity ($\dot{\theta}_{W,a}$) in allocentric coordinates of both left ($\theta_{W,aL}, \dot{\theta}_{W,aL}$) and right ($\theta_{W,aR}, \dot{\theta}_{W,aR}$) whisker beams (see "Methods" section and Supplementary Fig. 9). We then computed, for each neuron, the difference between its allocentric and whisker-centric MDs, for whisker angle and whisker angular velocity. We found that, as a population, VPM neurons were more sensitive to allocentric coordinates, whereas the population of POm neurons was more biased towards whisker-centric coordinates of whisking (Fig. 3c). This result is consistent with VPM neurons processing primarily object-related information and POm neurons processing primarily whisking-related information[1].

## The extent of thalamic coding of head kinematics

What is the extent of coding of head kinematics in the sensory vibrissal thalamus? The emergence protocol yielded relatively short durations of continuous recordings, with which weakly tuned neurons would not be revealed. Thus, the fractions of tuned cells presented in Fig. 2c should be taken as a lower bound. We thus designed an open field

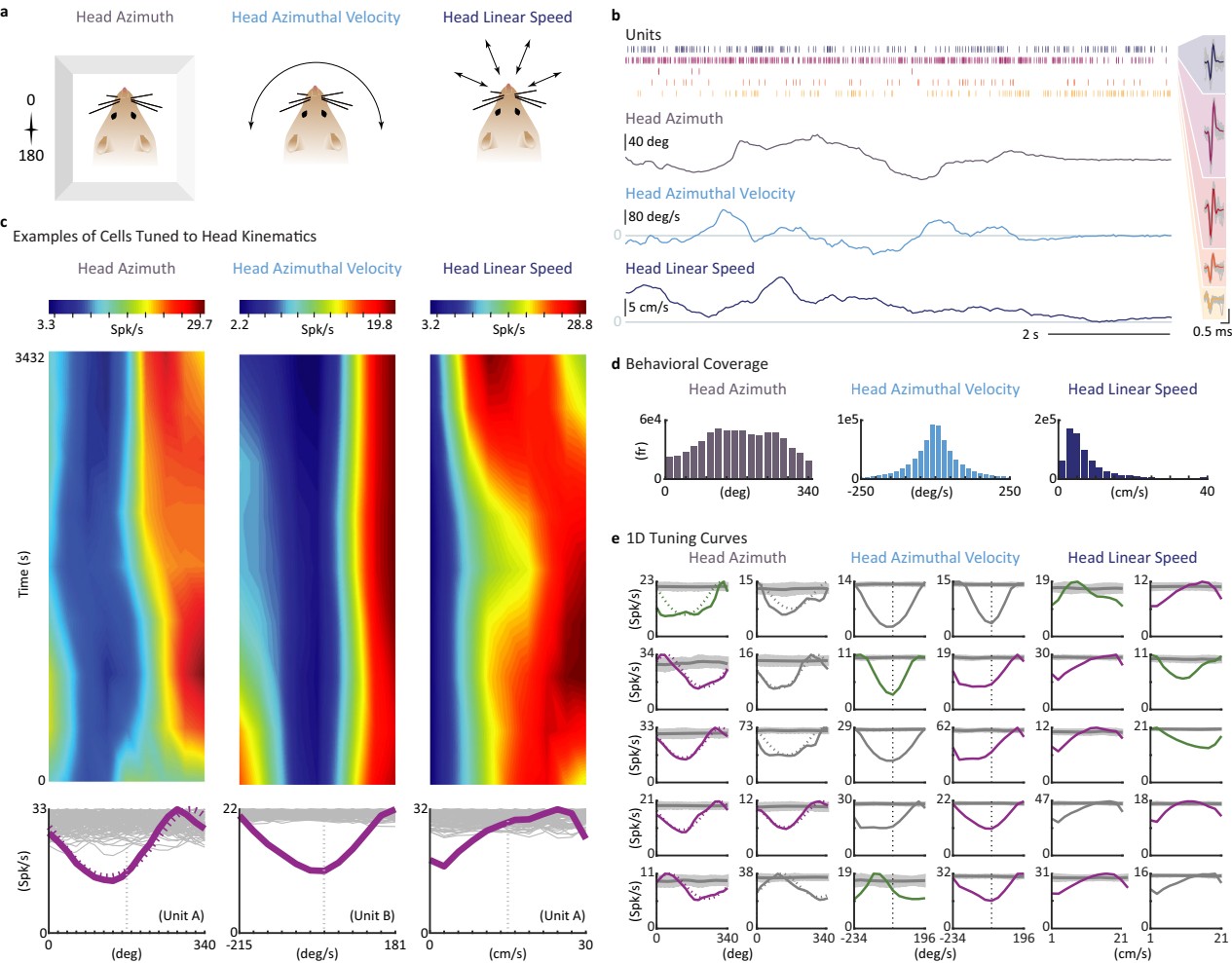

**Fig. 4 | Coding of head kinematics. a** Schematics of the three head kinematic variables analyzed in this study. **b** Simultaneous recordings of head kinematics (lower traces) and single-unit spike activity (raster plot, top colored traces). The spike shapes of the recorded units are depicted, using the same color code, on the right. **c** Examples of thalamic tunings to head kinematics. Tuning curves of two POm single units ("Unit A", left and right columns and "unit B", middle column), recorded in a whisker-trimmed mouse, are depicted. Top, tuning curves as a function of recording time (see "Methods" section for binning). Plots were smoothed. Bottom, mean tuning curves, averaged across the entire recording period. The gray traces depict 100 tunings computed from shuffled data (see "Methods" section). Each curve, of both empirical and shuffled data, is normalized to its maximal value; the firing rates labeling the ordinate in each plot refer to the empirical data only. **d** Distributions of the kinematic variables of the head throughout all recording sessions. **e** Examples of significant tuning curves computed for VPM, POm and untagged (gray) neurons. Gray traces are mean ± 1SD of the shuffled traces. Dotted curves depict the circular von Mises fits to head azimuth tuning (see "Methods" section). Examples were taken from 4 POm mice in 6 sessions (4 with whiskers and 2 without) and 3 VPM mice in 4 sessions (2 with whiskers and 2 without). Each curve, of both empirical and shuffled data, is normalized to its maximal value; the firing rates labeling the ordinate in each plot refer to the empirical data only. Source data are provided as a Source Data file.

experimental protocol that, unlike the emergence protocol, allowed long periods of head tracking (see "Methods" section). With this protocol, the behavior and thalamic activity of mice freely exploring an open arena were recorded during one ($n = 2$ mice) or two ($n = 5$ mice) continuous sessions of about 1 hour each[26]. The whiskers of the 5 mice that performed a second experimental session were trimmed to the fur level four hours prior to the beginning of the experimental session. During each session, the mice explored the arena continuously and extensively (Supplementary Fig. 6). In general, arena exploration did not exhibit statistically significant directional asymmetry: the distributions of head azimuth did not differ from a uniform distribution in 10/12 sessions ($p > 0.05$, permutation test). 174 neurons (27 POm-tagged and 63 VPM-tagged) were recorded in mice with untrimmed whiskers, and 130 neurons (11 POm-tagged and 22 VPM-tagged) were recorded in mice with trimmed whiskers. Since electrodes were not advanced between sessions, the neuronal populations recorded in mice that performed in two behavioral sessions likely partially overlapped.

A substantial proportion of tagged neurons (27/90 in 7 mice with untrimmed whiskers and 26/33 in 5 mice with trimmed whiskers) were tuned to at least one head kinematic variable: $\theta_H$, $\dot{\theta}_H$ or $V_H$ (Fig. 4a, b). Examples of well-tuned neurons are presented in Fig. 4c. In this example, two POm neurons, recorded simultaneously in a whisker-trimmed mouse, exhibited clear tuning to head azimuth and linear speed (unit A) and to azimuthal velocity (unit B). The tuning was stable throughout the entire recording session (3432 s) and, on average, relatively broad (Fig. 4c, bottom). Such broad tuning, exhibiting sensitivity to changes across significant portions of the behavioral range (Fig. 4d), was characteristic of most thalamic neurons (see examples in Fig. 4e). The modal values of VPM and POm neurons' tunings covered the entire behavioral range in intact mice (Fig. 5a).

Tuning strengths were also broadly distributed (Fig. 5b), with POm and VPM neurons exhibiting similar MDs for $\theta_H$, $\dot{\theta}_H$ and $V_H$ ($p > 0.15$, Mann Whitney $U$ test). In general, the tunings were stable. The tuning stability was quantified as the sample correlation coefficient

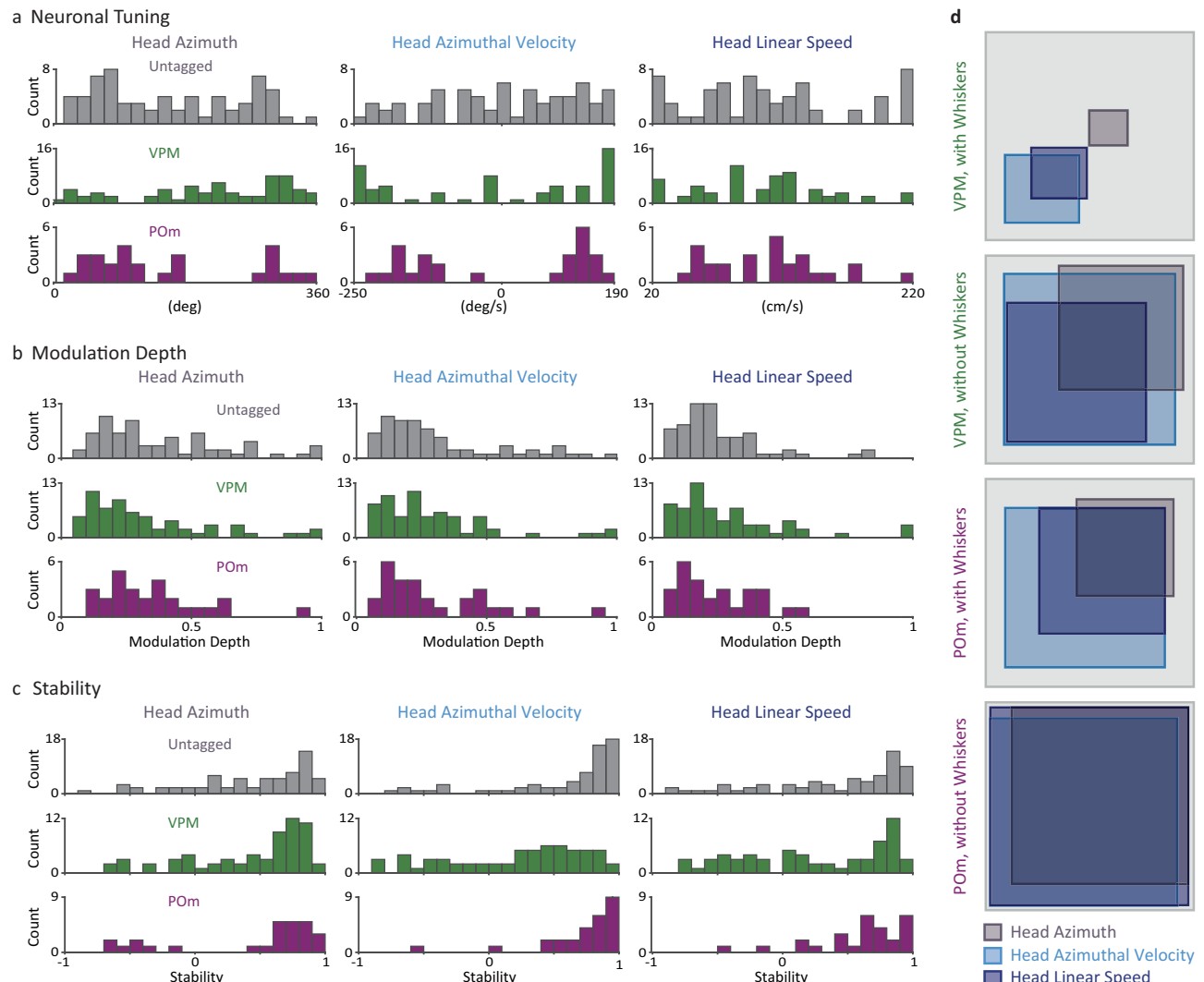

**Fig. 5 | Distributed coding of head kinematics across thalamic nuclei.**
**a** Distributions of tuning curves of all recorded neurons for the three kinematic variables (columns). Head kinematic azimuth angles were arbitrarily assigned values between 0 and 360 degrees within the symmetrical arena. **b** Distributions of modulation depths of all tuning curves shown in f. **c** Distributions of stability indices (see "Methods" section) for all tunings shown in f. **d** Venn diagrams of significant tunings for intact and whisker-trimmed neurons in VPM (top) and POm (bottom). Areas of regions are proportional to fractions of total counts. Source data are provided as a Source Data file.

$(r_{xy})$ between the tuning curves computed for each half of the recording session. The majority of tuned thalamic neurons exhibited stable tuning curves throughout the recording; yet, quite a few neurons exhibited either weak stability ($|r_{xy}| < 0.5$; 43% in VPM and 20% in POm, mean over the 3 kinematic variables) or even reversed tuning ($r_{xy} \le -0.5$; 11% in VPM and 5% in POm) along the session (Fig. 5c). In general, this finding indicates that coding of head kinematics is a major attribute of VPM and more so of POm neurons.

Whisker trimming increased the abundance of significantly tuned neurons both in POm (by 1.4 fold) and (more so) in VPM (by 4.6 fold; Fig. 5d). The increased abundance of significant tunings to head kinematics following whisker trimming indicates that sensitivity to head kinematics is not derived from sensitivity to whisker motion or to its re-afferent signals (re-afferent responses decrease with whisker trimming[27]).

## Thalamic influence on head kinematics

Anatomy suggests that POm and VPM neurons can affect motor functions via several pathways[11–17]. In order to test whether these thalamic neurons can affect head kinematics, we tracked mice in an open field arena while applying brief laser pulses of either 10 or 50 ms at frequencies of 2, 5, and 8 Hz. These laser pulses strongly activated large populations of neurons in the stimulated area, both via direct activation of the neurons expressing ChR2 and indirect activation of other interconnected neurons (Fig. 1). The responses of our recorded tagged neurons, which comprise a sample of these populations, probed the response profile of the activated population.

In several cases, we could measure the effects of individual laser pulses on head kinematics. For example, during a block of 12 trains of 2 Hz stimulations (2 s each, inter-train-interval = 2 s), changes in $V_H$ accurately followed the laser-induced changes in the activity levels of the tagged VPM neurons (Fig. 6a and Supplementary Movie 2). During this block, phase-locking increased from train to train, gradually approaching a one-to-one relationship between individual stimulation pulses, individual population response pulses, and (delayed) individual $V_H$ increments. As expected from a process in which changes in thalamic firing take part in changing head velocity, in this example the intensities of changes were correlated (Fig. 6b; $r^2 = 0.2$, $p = 0.001$) and neuronal changes tended to lead head velocity changes (Fig. 6c; $p = 0.009$, permutation test).

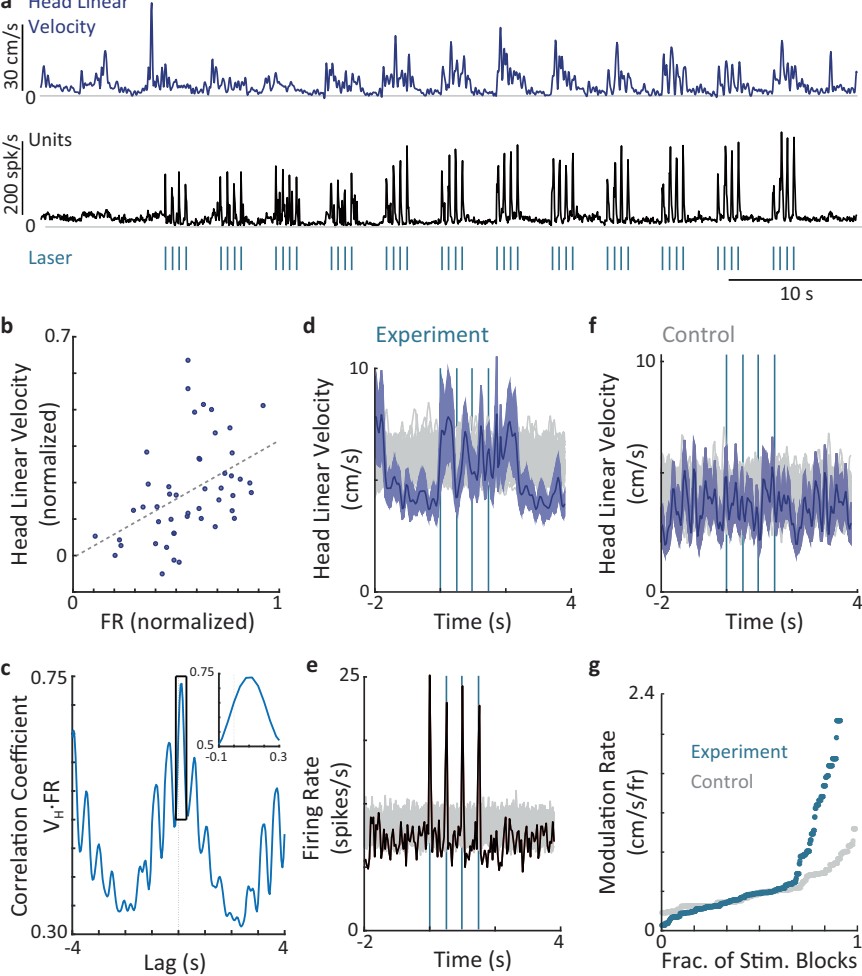

**Fig. 6 | Effects of thalamic stimulations on head kinematics. a** An example of a single block of laser stimulations at 2 Hz (12 repetitions, 2 s on 2 s off) in the VPM. Top, $V_H$. Middle, population firing rate of all simultaneously recorded tagged VPM neurons. Bottom, times of laser stimulations (175 mW/mm², 10 ms). **b** Head linear velocity versus firing rate for the data presented in **a**. Each data point refers to one laser pulse ($n = 48$). **c** Cross-correlation between the firing rate and head linear velocity presented in a. A correlation value at a lag L reflects, in a monotonous manner, the probability that changes in the linear velocity lag changes in the VPM firing rates by L. **d** An example of STA of $V_H$ of a VPM mouse, time-locked to the beginning of each train of 2 Hz stimulation ($t = 0$) within a single stimulation block

($n = 12$ trains). Mean (solid) and SEM margins (shaded) are depicted. Gray traces are STAs of shuffled data (100 repetitions). **e** STA of the mean firing rate of the tagged neurons recorded simultaneously with the kinematics shown in **d**. **f** Same experimental conditions as in **d** but with a control mouse in which fluorophore, but not ChR2, is expressed. **g** Modulation rates of $V_H$ obtained in experimental (green) and control mice, ordered by magnitude and up-sampled to equalize sample sizes; all the stimulation blocks conducted in the study are included (original samples: $n = 144$ experimental blocks and 64 control blocks). Source data are provided as a Source Data file.

The linear velocity of the head is directly related to the linear locomotion of the mouse (Supplementary Movie 2). The involvement of vibrissal neural networks in this process is expected from the tight links between vibrissal kinematics and head and body kinematics (Figs. 2a, 3b)[9]. While the anatomical pathways implementing these links need yet to be clarified, current knowledge suggests the projections of VPM neurons to deep layers of the somatosensory cortex[11,12], and the projections of POm neurons to the cortical and subcortical motor areas[13–15], as candidate pathways. Likewise, the pathways carrying head kinematic signals to the vibrissal thalamus are not yet known, with both afferent (somatosensory[28] and vestibular[29]) and cortico-thalamic[30,31] connections forming possible candidates.

Head kinematic changes induced by thalamic stimulations cannot be considered startle responses, as tactile startle responses are conveyed via brainstem trigeminal nuclei[32–34], which do not receive direct thalamic inputs. To control for the possibility that the light delivery affected head movements via pathways not involving the studied thalamic neurons (e.g., via neurons responsive to the light), control

mice expressing fluorophore but not ChR2 ($n = 4$ mice) underwent the same procedures as experimental mice and were recorded under the same conditions and stimulation protocols. We then compared the effects of laser stimulations on head kinematics using the modulation rate (MR, the mean modulation during stimulation, see "Methods" section). While significant $V_H$ modulations could be observed in experimental mice (e.g., Fig. 6d, e; note the periodic stimulus pattern at 0.5 Hz), only limited $V_H$ modulations were observed in control mice (e.g., Fig. 6f). A quantitative comparison of the two populations shows that $V_H$ modulations in control mice failed to reach the level demonstrated in 22% of the stimulation blocks in the experimental mice. The modulations in 30% of the experimental blocks were larger than those predicted by the modulations in control mice (Fig. 6g). Experimental modulations of $\dot{\theta}_H$ were typically weaker than those of $V_H$.

During free explorations, changes in $\dot{\theta}_H$ typically led those in $V_H$ by about 80 ms (Fig. 7a, black trace; Fig. 7b). Optogenetic stimulations changed these relationships in some cases, typically in the direction of decreasing the delay between the two variables (Fig. 7a, blue trace;

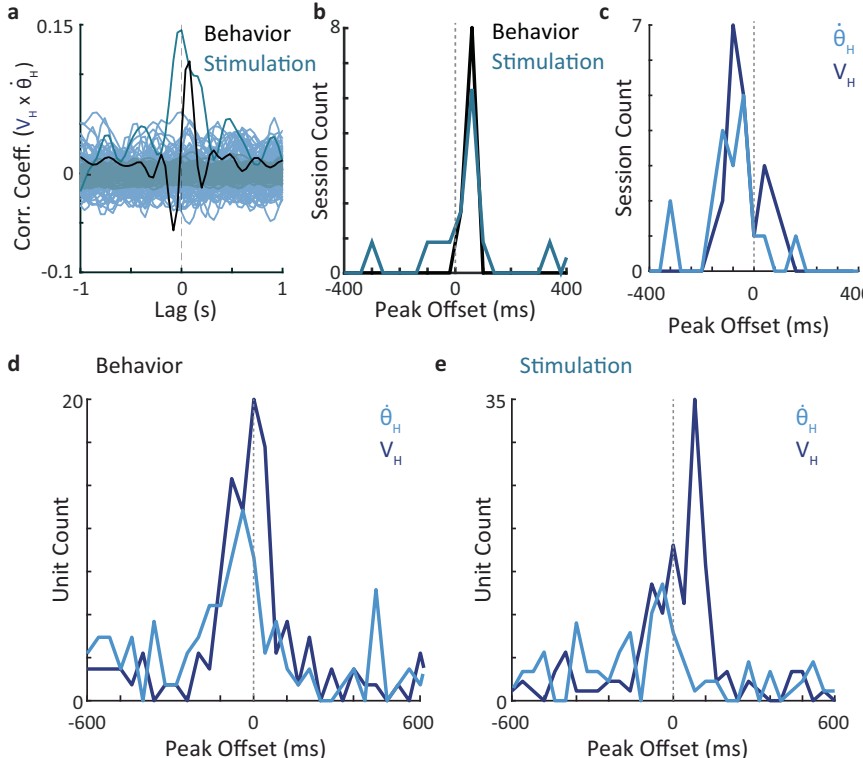

**Fig. 7 | Sensory-motor relationships of thalamic neurons. a** Cross-correlation functions between $\dot{\theta}_H$ and $V_H$ during free behavior (black) or laser stimulation (bluish). Low-intensity curves are cross-correlations between shuffled traces (100 repetitions). **b** Distributions of correlation peak delays between $\dot{\theta}_H$ and $V_H$ across recording sessions (11 free behavior and 17 stimulation sessions). **c** Distributions of correlation peak delays between neuronal population rates and $V_H$ (blue) or $\dot{\theta}_H$ (bluish) across all recording sessions ($n = 28$ sessions). **d, e** Distributions of correlation peak delays between firing rates of individual neurons and $V_H$ (blue) or $\dot{\theta}_H$ (bluish) across all tagged neurons recorded during free behavior (**d**) and stimulation (**e**) sessions. Source data are provided as a Source Data file.

Fig. 7b). Thalamic neuronal populations exhibited correlations whose peak delays typically led or lagged head kinematic changes by < 200 ms (Supplementary Fig. 7). In some cases, neuronal-head kinematic correlations could be modified by optogenetic stimulations, adding either sensory-like or motor-like neuronal activity (Supplementary Figs. 7b, d, respectively). When lumped together, the distribution of peak delays across all our experimental mice exhibited a distribution that spanned zero delay, in which, in most cases, thalamic neurons lagged kinematic changes, and in a minority of the cases, they led them (Fig. 7c). This suggests a scheme in which thalamic neurons of the vibrissal sensory nuclei participate in functional loops that affect head kinematics[9].

Analysis of peak delays for individual neurons revealed a more detailed picture. During free exploration, activity changes in most neurons lagged changes in $\dot{\theta}_H$ and in $V_H$ (Fig. 7d). The relative fraction of neurons that lead changes in $\dot{\theta}_H$ and $V_H$ significantly increased with optogenetic stimulations (Fig. 7e); for delays between −200 ms to 200 ms, the fraction increased from 30% to 47%. For reasons that are not yet clear to us, the effect was most evident with $V_H$, for which the fraction of leading neurons increased from 35% to 58% upon stimulation (Supplementary Fig. 8). The increase in prevalence of positive peak delays (between 0 and 400 ms) was statistically significant for $V_H$ ($p = 0.02$) and not for $\dot{\theta}_H$ ($p = 0.16$) (Permutation tests, see "Methods" section). It is important to mention that, while being effective, the spikes induced by our optogenetic stimulations did not over dominate neuronal activity, and thus likely did not cancel the sensory bias observed before the stimulation. During anesthesia, the number of spikes evoked by laser stimulations in each tagged unit comprised, on average, $0.52 \pm 0.29$ (mean ± SD) of the spikes produced by that neuron during the stimulation period (see "Methods" section). During behavior, laser stimulations increased the mean firing rate of tagged

units by $15.5 \pm 14.5$ % (mean ± SD), on average for the entire stimulation period.

It is possible that the observed shift in relative timing between neuronal activity changes and head kinematic changes reflect two unrelated processes, both induced by the stimulations. However, the observed delays between neuronal and kinematic changes (typically between 40 and 120 ms) are longer than those characterizing brainstem-based startle responses[34] and shorter than those characterizing cortex-based perceptual reports[35]. In contrast, these delays are consistent with the possibility that the VPM and POm, considered thus far to focus on the processing of sensory vibrissal information, also play an active role in sensory-motor loops of head kinematics. Yet, it is important to note that artificial stimulations of neuronal groups can induce activity states that do not occur naturally and can activate pathways that are not activated during natural behavior[36].

## Discussion

The development of experimental tools over the years induced a rather dramatic change in our conception of the roles of sensory thalamic nuclei in brain function. Thus, the early conception that these nuclei are analogous to passive relay stations in a communication channel (in this case, from sensory organs to the cerebral cortex) was repeatedly shaken by accumulating results such as those indicating active functions of thalamocortical loops, intra-thalamic processing and brainstem-thalamic code transformations (reviewed in refs. 17,37). Our current study adds an unexpected observation: the vibrissal sensory thalamus integrates head motion and whisking information.

The tuning to head kinematics in the sensory vibrissal thalamus was broad, in contrast to the sharp tuning of head azimuth in the motor thalamus[38], and similar to frequently reported cortical tunings[39,40]. This suggests that vibrissal thalamic neurons do not code

head kinematics per se, and their firing cannot contribute much to circuits that need to use exact head kinematics. Instead, it seems that head kinematics plays a significant role in the processing of these neurons. This suggestion is tenable given the tight coordination between head and whisker movements[9,41] and the crucial dependency of object localization on both[1].

Recording from POm and VPM neurons in freely moving mice also provides a possible resolution to a long-standing puzzle in vibrissal processing. Previous studies revealed varying relations between POm and VPM responses to whisking, depending on the experimental context, resulting in continuing debates about whether POm neurons are involved in whisking processing[4,6–8]. On the other hand, it was recently shown that the experimental context is crucial: using head fixation or whisker trimming significantly affects how rodents control their whisking[9]. The current study shows that when mice are free to move in an arena, both POm and VPM neurons participate in the processing of whisking in a comparable manner. More specifically, our analysis suggests this processing is biased towards whisker-centric coordinates in the POm and allocentric coordinates in the VPM, which is consistent with the VPM specializing in the processing of external objects and POm in the processing and control of self-motion[4,8].

Why is POm silenced in head-fixed rodents[6,7]? From a functional point of view, this may reflect the cancelation of head motion, implying that in awake animals, POm activity depends on both head and whisker kinematics. Why, then, is POm not silenced in anesthetized rodents? Answering this question should provide valuable information about how head kinematics activates POm neurons. This mechanism is likely based on the loop connecting the POm, motor cortex, and zona incerta[42,43]. In normal conditions, this loop has been hypothesized to participate in tuning the motor-sensory control of whisker kinematics, such as whisking frequency, in the POm[5,17]. The current results suggest that this control, if taking place in mice, disables POm activation when the head is immobile in awake conditions. This is consistent with the observation that inhibition of primary motor cortex in head-fixed mice increases POm correlation with whisking[44]. Such a control mechanism might be related to the freezing behavior of rodents under threat, during which whisking freezing is required to follow head freezing.

Overall, our results are consistent with the dynamical view of perception, in which perception gradually emerges from iterative processes along closed motor-sensory-motor loops. Our observations support the notion that as iterative processing strives to converge to a steady-state of sensor kinematics and sensation, head and whisker movements should be processed in tandem. Our recordings within the sensory thalamic nuclei VPM and POm provide compelling evidence for this co-coding phenomenon. Furthermore, the finding that co-coding occurs in both nuclei underscores the evolutionary significance of concurrent head-whisker processing, suggesting its enduring importance across different evolutionary stages – in this case, those involving the evolution of the paralemniscal and lemniscal systems. While our study sheds light on the co-coding aspect, further investigations are warranted to unravel the specific dynamical processes involved in object perception within these nuclei, to elucidate the division of labor between them and to clarify their potential effects on head and whiskers kinematics.

## Methods
### Animal subjects
All animal experiments were performed according to protocols approved by the Weizmann Institute's Institutional Animal Care and Use Committee (IACUC). Male mice (1.5 – 5 months old; males were used since this was the common practice in the field when we started this project) were housed in groups prior to surgery and then single-housed after surgery to avoid damage to electrical and optical implants. Mice were maintained on a 12-hour light/dark diurnal cycle and were given *ad libitum* access to food and water, except while performing behavioral sessions. During behavioral sessions, mice were maintained on a reversed 12-hour light/dark diurnal cycle. Data were acquired using the Gpr26-Cre mouse line, a line of transgenic mice heterozygous for the *cre* transgene, which selectively expresses the Cre recombinase in the POm, and shows sparse expression in the VPM[18,45]. Gpr26-Cre mice were crossed with Ai9 mice to verify Cre expression patterns. Gpr26-Cre heterozygous mice were also used to confirm the efficacy of the Cre-ON and Cre-OFF viral constructs (Supplementary Figs. 1 and 2; see also Viral Injections).

### Chronic multisite optrode construction
Chronic multi-site optrodes (CMOs) were constructed in-house, following an adaptation of the protocol developed by Yizhar et al.[19]. CMOs consisted of an array of fourteen 25 μm-diameter tungsten wires (WireTronic), precut to a length of 5 cm, adhered to an implantable fiber-optic light guide (Doric Lenses). The light guide consisted of a 200 μm-core optical fiber fed through a 1.25 mm ID metal ferrule. The optical fiber was shortened to a length of 8 mm from the base of the metal ferrule, and the electrode array was cut so that the electrodes tips lay within a 0.5 mm from the tip of the fiber. The fiber-electrode bundle was passed through a 5 mm long 320 μm ID 25-gauge polyimide guide tube, which was then attached with cyanoacrylate glue to a plastic PVC plate machined in-house to 12 mm length x 1.25 mm width x 4 mm height with a groove of 4 mm length × 1.25 mm width × 1 mm depth centered in the middle of the plate. The fiber-electrode bundle was fit into the PVC plate groove so that the longest electrode extended 7 mm from the base of the plate. The plate was then affixed to a Mill-Max PCB connector also using cyanoacrylate glue. The electrode wires were threaded through the holes in the Mill-Max connector and connected using gold pins. Two stainless steel 125 μm-diameter wires (A-M Systems) were cut to a length of 7 cm and were connected to the Mill-Max connector with gold pins. These wires served as reference and ground wires. Epoxy (Devcon Home 5 Min Epoxy) was used to cover all exposed parts of the electrode wires (Supplementary Fig. 3e).

CMOs for control experiments were constructed in a similar manner, but did not contain electrodes.

### Viral injections, CMO, and optic fiber implantation
Recombinant AAV was obtained from the University of North Carolina Vector Core (Chapel Hill, North Carolina, USA) or made in-house. Mice were anesthetized with 4% isoflurane and mounted in a stereotactic device (Kopf Instruments), and then maintained under 1%-1.5% isoflurane for the duration of the surgery.

**Viral injections for efficacy test of Cre/ON-Cre/OFF targeting strategy.** Lambda and bregma were leveled, and a craniotomy was performed in which a disc-shaped piece of skull approximately 2 mm in diameter was removed above the whisking-related thalamus (from bregma: 1.50 medial on the left side and 1.82 caudal). Then, 0.5 μl of AAV2/1 virus with Cre-OFF dependent expression (double-floxed open reading frame, DFO) of ChR2-EYFP under the control of a human synapsin (hSyn) promoter (AAV2/1.hSyn. DFO.ChR2-EYFP, 2E + 12 vg/ml), and 0.5 μl of AAV8 virus with Cre-ON dependent expression (double-floxed inverse open reading frame, DIO) of hM4D-mCherry under the control of a human elongation factor-1α (EF1α) promoter (AAV8. EF1α. DIO.hM4D-mCherry, 8E + 12 vg/ml) mixed 1:1 by volume were injected to the whisking-related thalamus of GPR26-Cre mice (from bregma: 1.50 mm medial on the left side, 1.82 mm caudal, and 4.00 mm ventral) at a rate of 100 nL/min using a syringe pump (World Precision Instrument) and a 10 μL syringe with a 34 gauge beveled needle (World Precision Instruments). After allowing the virus ten minutes to diffuse, the needle was removed, the incision was adhesively closed (VetBond, 3 M) and the mouse was removed from the stereotactic device.

**Viral injections and CMO implantation for anesthetized-mouse and behavioral sessions.** 1.0 μl of either AAV2/1 virus with Cre−OFF dependent expression of ChR2-EYFP under the control of an hSyn promoter (VPM mice, AAV2/1.hSyn. DFO.ChR2-EYFP, 2E + 12 vg/ml, Supplementary Fig. 3b, d), or AAV2/1 virus with Cre-ON dependent expression of ChR2-EYFP under the control of an EF1α promoter (POm mice, AAV2/1. EF1α. DIO.ChR2-EYFP, 4E + 12 vg/ml, Supplementary Fig. 3a, c) was injected to the whisking-related thalamus of GPR26-Cre mice following the same protocol described above[45].

The CMO was implanted at least two weeks after the injection of virus to allow for expression of ChR2 prior to the implantation surgery. Mice were anesthetized with a ketamine - xylazine cocktail (160 μg ketamine/g body weight, 20 μg xylazine/g body weight) and mounted in a stereotactic device (Kopf Instruments). Lambda and bregma were leveled, and a craniotomy was performed in which a disc-shaped piece of skull approximately 3 mm in diameter was removed caudal to the whisking-related thalamus (mice with electrode tips in the VPM – from bregma: 2.00 mm medial - left and 3.79 mm caudal; mice with electrode tips in the POm – from bregma: 1.50 mm medial - left and 3.67 mm caudal). The CMO was then lowered at a 30° angle from the surface of the cranium (mice with electrode tips in the VPM – from bregma: 2.00 mm medial - left side and 3.85 mm caudal; mice with electrode tips in the POm – from bregma: 1.50 medial - left side and 3.73 mm caudal). While the CMO was lowered into the brain, neuronal responses to 447 nm blue laser (PSU-III-LED, OEM Laser Systems) and to whisker stimulation via air puff delivered through a solenoid pinch valve (ASCO Series 284-S, ASCO Scientific) were monitored using electrophysiology (Open Ephys platform). The CMO was stereotactically lowered until neuronal responses to both laser stimulation and whisker stimulation were observed, typically between 3.20 mm and 4.00 mm ventral from bregma in VPM mice, and 3.00 mm and 4.00 mm ventral from bregma in POm mice. The ground and reference wires of the CMO were each wrapped around a metal screw inserted through the caudal part of the skull, over the cerebellum. To ensure the stability of the implant, luting cement was applied to the skull (C&B Metabond). The CMO was then cemented to the mouse's skull using dental acrylic. After the dental acrylic dried, the incision was closed with 3 M Vetbond Tissue Adhesive and the mouse was removed from the stereotactic device. Post-surgery, mice were closely monitored and treated with analgesic (Buprenorphine, 5 μl/g of body weight, S.C.) and antibiotics as needed.

**Viral injections and optic fiber implantation for control sessions.** General anesthetic and surgical procedures were performed as described above. A craniotomy was performed above the whisking-related thalamus of GPR26-Cre mice (from bregma: 1.50 medial on the left side and 1.82 caudal). Then, 0.5 μl of AAV2/1 virus with Cre−ON dependent expression of mCherry under the control of an EF1α promotor (AAV2/1. EF1α. DIO. mCherry) was injected to the whisking-related thalamus (from bregma: 1.50 mm medial on the left side, 1.82 mm caudal, and 4.00 mm ventral). A CMO without electrodes was implanted during the same surgery: from bregma: 1.50 mm medial on the left side, 3.73 mm caudal and lowered in 30° angle 3.6–4 mm.

**Optogenetic and electrophysiological apparatus**
**Apparatus for sessions with anesthetized mice.** All anesthetized-mouse experiments were performed in a well-lit, quiet room. The body temperature of each anesthetized mouse was maintained in each session by placing it on an activated, heat delivering, paper-towel covered SpaceGel pad (Braintree Scientific) on a 12.5 cm length × 8.5 cm width × 4 cm height block with its head and upper torso extended beyond the block. The block was placed on a clear, transparent polycarbonate plate (20 cm in length, 18 cm in width) that was held 15 cm above the surface of a table. The polycarbonate plate was backlit by an

IR-LED array (880 nm wavelength, 23 cm × 23 cm, Metaphase, USA). A high-speed camera (1280 × 1024 pix, 500fps, CL600x2/MFM, Optronics; with Karbon PCI express frame grabber, BitFlow) filmed the experimental area from above (Supplementary Fig. 4). Recording of both the high-speed camera data by a RAID computer and the electrophysiological data by a Neuralynx Digital Lynx 4SX 32-channel system were manually triggered by the user via an in-house program (written by Dr. Enrico Segre) when stimuli were delivered to the anesthetized mice. Video and electrophysiological recording were manually stopped after the delivery of the stimuli. Two IR-LEDs (940 nm wavelength) were placed within the field of view of the high-speed camera. One IR-LED was triggered to emit light when the laser stimulus was being delivered, and the other IR-LED was triggered to emit light when the air puff stimulus was delivered. Additionally, the LEDs could be triggered through Matlab (Mathworks) for data synchronization purposes in cases in which neither laser nor air puff stimuli were delivered.

The laser stimulus (460 nm, Omicron NanoTechnology) was delivered through a fiberoptic patch cord. The 460-nm laser was driven directly by a Matlab (MathWorks) program through a DAQ board (National Instruments,USB-6353).

The air puff (compressed air) stimulus was delivered via rubber tubing fed through a solenoid pinch valve (ASCO Series 284-S, ASCO Scientific) that was triggered to open and close to control the flow of the compressed air. A micropipette tip (0.38 mm diameter opening) was placed at the end of the rubber tubing to direct and narrow the flow of the compressed air. The micropipette tip was positioned between 5 mm and 10 mm from the whisker pad. The solenoid valve was driven directly by a Matlab (MathWorks) program through the NI DAQ board. The electrophysiology system used in the experiments was a Neuralynx Digital Lynx 4SX 32-channel system running Cheetah software with an electrical-optical commutator. Recorded neuronal signals were amplified using a HS-18-CNR-light-emitting diode unity-gain headstage amplifier, filtered (600–6000 Hz) and digitized at 32 kHz. Spike sorting was performed offline (see Data Analysis and Statistics).

**Apparatus for emergence sessions (OptoWhisk).** All behavioral experiments except for the Open Field sessions were performed using a behavioral apparatus, the OptoWhisk, which was custom designed for these experiments. Behavioral experiments were performed in a darkened, quiet room[9,26]. The OptoWhisk consisted of a holding cage (20 cm length, 26 cm width, 23 cm height) that had a sliding door (12.4 cm width, 23 cm height) that could be opened to allow an animal to emerge into the experimental area (Supplementary Fig. 4). Both the holding cage and the experimental area were held 15 cm above the surface of a table. The experimental area consisted of a clear, transparent polycarbonate plate (18 cm in length and 17 cm in width), bordered by three transparent polycarbonate walls (18 cm in length and 17.5 cm in height). A polycarbonate cylinder (2.5 cm diameter, 6 cm height) was placed in the experimental area. The location of the object was manually changed between experiments. The polycarbonate plate was backlit by an IR-LED array (880 nm wavelength, 23 cm × 23 cm, Metaphase, USA). A high-speed camera (as described above) filmed the experimental area and custom-written software was used to trigger the collection of both the high-speed camera data and the electrophysiological data whenever the mouse emerged from the holding cage into the experimental area. Video and electrophysiological recording stopped when the mouse returned to the holding cage. Additionally, two IR-LEDs (940 nm wavelength) were placed within the field of view of the high-speed camera. One IR-LED was triggered to emit light when the laser stimulus (460-nm laser, Omicron Nano-Technology) was being delivered. The second IR-LED constantly emitted 10 ms pulses of light at 5 Hz to allow synchronization of the electrophysiological and behavioral data. Neuronal spiking and local

field potential data were recorded during experiments. Spike sorting was performed offline (see Analysis subsection).

**Apparatus for open field sessions.** The Open Field behavioral sessions were performed in a darkened, quiet room. The Open Field consisted of a large, cubic box (50 cm side) with an open top (Supplementary Fig. 5). In this experiment, a red (635 nm) and a green (565 nm) LED were inserted into either side of the Neuralynx headstage used to record electrophysiological data. These LEDs enabled the online tracking of head position and angle. The Open Field was filmed from above using an analog camera compatible with Cheetah video tracking software (720 × 576 pix, 25 fps, CV-S3200, JAI). Video data was automatically synchronized with electrophysiogical data in the Neuralynx Digital Lynx 4SX 32-channel system. Using Cheetah, both neuronal spiking and local field potential data were recorded during experiments. Spike sorting was performed offline (see Analysis subsection).

**Behavioral, optogenetic, and electrophysiological sessions**
**Anesthetized-mouse Sessions.** The subjects of this experiment were 4 Gpr26-Cre heterozygous mice in which ChR2-EYFP was expressed in either the VPM (VPM25 and VPM28) or the POm (POm31 and POm36) using the Cre-ON/Cre-OFF genetic targeting strategy, and which were implanted with CMOs as described above.

The experiment consisted of recording the response of thalamic units to air puff and laser stimulation. Prior to the experiment, mice were anesthetized using a ketamine - xylazine cocktail (80 µg ketamine/g body weight, 10 µg xylazine/g body weight). Anesthetized mice were connected to the Neuralynx DigitalLynx system with a headstage and to a 460-nm blue laser with an optical fiber, and then placed under a high-speed, high-resolution camera so that the whiskers and upper torso of the mice were visible to the camera. For the duration of the application of all stimuli, the camera and DigitalLynx system were manually triggered to enable the recording of video and electrophysiological data.

Laser stimulation (460 nm blue; 8.5–400 mW/mm$^2$) was first delivered in blocks of 12 repetitions of a 2 s, 2 Hz train of 10 ms laser pulses (4 pulses per train) with 2 s between train repetitions. This was followed by a block of 12 repetitions of a 2 s, 5 Hz train, then a block of 8 Hz trains, and then another block of 2 Hzs train of 10 ms laser pulses (10 pulses, 16 pulses, and 4 pulses per train, respectively) with 2 s between train repetitions. There were 8 s between blocks of different frequencies. The same stimulation protocol was then repeated with 50 ms pulses. Air puff stimulation was then delivered to the contralateral whiskers with the same stimulation protocol as was used for laser stimulation so that a direct comparison of responses could be made.

During air puff and laser stimulations, an IR-LED within the field of view of the camera emitted light during the duration of the 10 ms or 50 ms air puff and laser pulses. During all periods in which neither laser nor air puff stimulation was delivered, an IR-LED delivered 10 ms light pulses at 5 Hz within the field of view of the camera to enable synchronization of video and electrophysiological data.

**Emergence sessions.** The subjects of this experiment were 10 Gpr26-Cre heterozygous mice in which ChR2-EYFP was expressed in either the VPM (4 mice) or the POm (6 mice) using the Cre-OFF/Cre-ON genetic targeting strategy, respectively, and which were implanted with CMOs as described above.

An Anesthetized-mouse Session preceded each Emergence Session. After the Anesthetized Session, once the mouse recovered from anesthesia enough that it was moving its limbs, the mouse was placed in the holding cage of the OptoWhisk apparatus while it was still connected to the DigitalLynx. The mouse was then left for four to six

hours in a darkened, quiet room to completely recover from anesthesia before the mouse performed the Emergence Session.

An experimental session consisted of recording an animal's whisking, head motion, and locomotive behavior. At the beginning of the Emergence Session, the sliding door blocking the exit of the holding cage was removed and the animal was free to leave the holding cage and explore the experimental area at will. As described in the Apparatus subsection, electrophysiological data was only recorded when the mouse was in the experimental area. The experimental period lasted for two to six hours, during which time a pseudorandom 460 nm laser stimulus (400–1200 mW/ mm$^2$; 2 s train of 10 ms laser pulses delivered at a randomly assigned frequency of either 2 Hz, 5 Hz, 8 Hz, or 16 Hz (4, 10, 16, or 32 pulses per train, respectively) with a pseudorandom interval of 15–45 s between trains) was delivered through the implanted fiberoptic light guide. At the end of the experimental session, the animal was removed from the behavioral apparatus, disconnected from the fiberoptic patch cord and the headstage, and placed in its home cage. The behavioral apparatus was then cleaned in preparation for the next animal. Whisker-trimmed mice performed the emergence task in the same manner as mice with untrimmed whiskers.

**Open field session (with whiskers).** The subjects of this experiment were 7 Gpr26-Cre heterozygous mice in which ChR2-EYFP was expressed in either the VPM (3 mice) or the POm (4 mice) using the Cre-OFF/Cre-ON genetic targeting strategy, respectively, and which were implanted with CMOs as described above.

The Open Field session was performed in a darkened, quiet room. In this test, an awake mouse was connected to a fiberoptic patch cord and a headstage, and then allowed to habituate in its home cage for about 30 min. While the mouse was habituating, laser stimulation was delivered so that laser-responsive units could be identified. The 460-nm laser stimulation was first delivered in a block of 12 repetitions of a 2 s, 2 Hz train of 10 ms laser pulses (4 pulses per train, 460 nm, 800–1200 mW/ mm$^2$) with 2 s between train repetitions. This was followed by a block of 12 repetitions of a 2 s, 5 Hz train, then a block of 8 Hz trains, and then a second block of 2 Hz trains of 10 ms laser pulses (10 pulses, 16 pulses and 4 pulses per train, respectively) with 2 s between train repetitions. There were 8 s between repetitions of different frequencies. The laser stimulation protocol was then repeated with 50 ms pulses. This is the same protocol used in the anesthetized recording experiment described above. After the mouse had habituated to the fiberoptic patch cord and headstage (about thirty minutes), the mouse was moved to the Open Field. The mouse was then allowed to explore the Open Field at will for approximately an hour while electrophysiological and video data were recorded. In one mouse, a second round of photostimulation, identical to that delivered in the homecage during habituation, was delivered at the end of the experiment. The mouse was then disconnected from the fiberoptic patch cord and the headstage, and returned to its home cage. The Open Field was then cleaned in preparation for the next animal.

**Open field session (trimmed whiskers).** The subjects of this experiment were 5 Gpr26-Cre heterozygous mice in which ChR2-EYFP was expressed in either the VPM (2 mice) or the POm (3 mice) using the Cre-OFF/Cre-ON genetic targeting strategy, respectively, and which were implanted with CMOs as described above. The mice were lightly anesthetized with 1%-1.5% isoflurane prior to whisker trimming. All whiskers were trimmed to the level of the fur. Mice performed in the Open Field Session approximately four hours after their whiskers were trimmed. The Open Field Session was performed as described above, except that in all mice, a second round of photostimulation, identical to that delivered in the homecage during habituation, was performed at the end of the experiment.

**Open field control sessions.** The subjects of this experiment were 4 Gpr26-Cre heterozygous mice in which mCherry was expressed in the whisking-related thalamus using the Cre-ON/Cre-OFF genetic targeting strategy, and which were implanted with CMOs, as described in the Viral Injections and Optic Fiber Implantation for Control Sessions section. An Open Field experiment was performed as described above in which the mouse first habituated in its homecage and then was allowed free exploration in the Open Field. Neuronal activity was not recorded in these control experiments. Photostimulation (460 nm, 800 to 1200 mW/ mm²) was delivered to these animals in both the homecage and the open field apparatus in a manner similar to that described above.

### Histology

Electrolytic lesions were performed in all mice implanted with CMOs. Prior to the lesion, mice were anesthetized with ketamine · xylazine cocktail (160 µg ketamine/g body weight, 20 µg xylazine/g body weight). One electrode on which high-quality electrophysiology recordings were obtained was chosen to deliver the electrolytic lesion to the brain. In the electrolytic lesion, 100 µA of electrical current was delivered via the chosen electrode to the brain for five seconds with positive polarity, and then for another five seconds with negative polarity. Twenty minutes after lesions were performed, mice were perfused as described below.

Animals were anesthetized with Pentobarbital (1 µl/1.5 g body weight; I.P.), and then perfused transcardially (4% paraformaldehyde in 0.1 M phosphate buffer, pH 7.2, cold). The brain was then removed and stored in PFA solution for twenty-four hours. The PFA solution was then replaced with 30% sucrose in 0.1 M phosphate buffer. After two to three days, brains were sliced coronally with a freezing microtome (Leica) to obtain sections with a thickness of 35 µm. Slices were stored in a cryoprotectant solution (25% glycerol, 30% ethylene glycol, in 0.012 M phosphate buffer at pH 6.7). Some brain slices were then stained for DNA using 4′,6-diamidino-2-phenylindole (DAPI). After the slicing and staining procedures, brain slices were mounted on slides, and were then visualized and imaged using a confocal microscope (Zeiss Axiovert 700).

### Whisker and head tracking, spike sorting and optogenetic tagging

Whisker and snout tracking of data collected in the emergence sessions was performed using the BIOTACT Whisker Tracker (BWT)[9,22]. This tracker provided the mouse's head position and head angle in each video frame through calculations based on the location of the mouse's nose tip and the center of its snout. Additionally, the tracker identified whiskers and provided the base angle of each of the tracker-identified whiskers with respect to the skin of the mouse's whisker pad. Median angle whisking beam traces were obtained by taking the median of all the base angles given by the BWT of all the identified whiskers on each side of the mice's snouts. These raw traces were then band-pass filtered between 8-35 Hz.

Extracellularly recorded signals were filtered (600-6000 Hz) at the time of recording (see Optogenetic and Electrophysiological Apparatus). In order to reduce global electrical noise artifacts, using a custom Matlab script, the mean signal from all simultaneously recorded channels was calculated and subtracted from each individual channel, and sorted into spikes belonging to single- and multi-unit clusters using the Plexon OfflineSorter 3.2.4 (Plexon), based on a PCA of waveform characteristics. The criteria for single-unit clusters were a clear separation from other clusters and a negligible number of short (<1 ms) inter-spike intervals (<0.1%; this value should not be confused with the false positive rate of spike detection). Only spikes belonging to single-unit clusters were analyzed in this study. Laser, kinematic, video and neuronal data were synchronized based on simultaneous input to all systems from TTL-triggered LEDs, as described in the Optogenetic and Electrophysiological Apparatus section.

Identified neuronal units were affiliated with the POm, the VPM, or neither based on an optogenetic tagging procedure using response magnitude and latency. Specifically, the mean firing rate of a unit in the 100 ms after laser stimulus onset was compared to the baseline firing rate of the unit in the 2 s prior to the beginning of the stimulus train (see Anesthetized-mouse and Open Field Sessions). The response Z-score for each unit was calculated across repetitions of laser stimulation, and units with Z-score ≥ 0.5 were considered potentially tagged units[46]. Tagging procedures were conducted during anesthesia, during each emergence session and immediately before each open field session.

The onset latency of the units' responses to laser pulses was determined using poststimulus time histogram (PSTH) analysis as the latency in which the PSTH crosses 10% of its maximal firing probability. A potentially tagged unit was defined as a "tagged unit" if its onset latency was < 6 ms (see Fig. 1). During the tagging procedure, the number of spikes evoked by laser stimulations in each tagged unit comprised, on average, 0.52 ± 0.29 (mean ± SD) of the spikes produced by that neuron during the stimulation period. Tagged units were affiliated with either POm (in Cre-ON mice) or VPM (in Cre-OFF mice).

### Samples sizes and mice allocation

Sample sizes were determined heuristically, based on accumulated experience, as the minimal sizes allowing conclusive results. Mice were randomly assigned to experimental groups.

### Data analysis and statistics

**Neuronal tuning curves for head and whisker kinematics.** Three head kinematic variables were analyzed: head azimuth in allocentric coordinates ($\theta_H$), head azimuthal velocity ($\dot{\theta}_H$) and head linear speed ($V_H$). Five whisker kinematic variables were analyzed for the median angle of the whisker beam on each side of the snout (see Whisker, Snout and Body Tracking, Spike Sorting and Analysis): offset ($\theta_{off}$), angle ($\theta_W$), angular velocity ($\dot{\theta}_W$), phase ($\varnothing_W$) and frequency ($\dot{\varnothing}_W$). Two allocentric kinematic variables were analyzed for the median angle of the whisker beam on each side of the snout: azimuth ($\theta_{W,aL}$, $\theta_{W,aR}$) and azimuthal velocity ($\dot{\theta}_{W,aL}$, $\dot{\theta}_{W,aR}$) in allocentric coordinates. These allocentric whisker variables were calculated by summing the concurrent corresponding head and whisker variables. Tuning curves for each variable and unit were computed by binning the kinematic variable into 18 (for head azimuth) or 12 bins, and computing the firing rate in each bin by dividing the number of spikes in that bin by the total duration of time the mouse's behavior belonged in that bin. The range of each variable was computed for each mouse from its behavioral trajectory. In all mice, head azimuthal angles (0 to 360 degrees) were sufficiently sampled so the entire range was used for each mouse's tuning range (18 bins of 20 degrees each); a von Mises circular normal function was also fitted to each such tuning[40]. The ranges of the other variables varied slightly from mouse to mouse, and the tuning curves were computed accordingly.

For consistency with the analysis of hippocampal head-direction cells in prior literature[40], the directionality of the azimuthal tuning curve was quantified by computing the Rayleigh vector length of the circular distribution, using the following equation:

$$\text{Rayleigh vector length} = \left( \frac{\pi}{n \sin\left(\frac{\pi}{n}\right)} \right) \cdot \frac{\sum_{j=1}^{n} r_{\varphi_j} e^{-i\varphi_j}}{\sum_{j=1}^{n} r_{\varphi_j}} \tag{1}$$

where

$n$ is the number of circular head-direction bins, $\varphi_j$ is the direction in radians of the $j$-th circular bin (namely, $2\pi j/n$), and $r_{\varphi j}$ is the average firing rate given the animal's head direction.

For each tuning curve, we applied a shuffling-based significance test. For each recorded neuron, the entire sequence of spikes was shifted in time by a uniformly distributed random time interval which was less than the duration of the recording with the end of the shifted sequence wrapped to the beginning of the original one. Using this procedure, the spike count and the temporal structure of the neuron's firing were preserved, but dissociated from the animal's actual behavior. This procedure was repeated 100 times for each tuning curve of each neuron. A tuning was considered statistically significant if its modulation depth exceeded the 95th percentile of the shuffled distribution for this variable and neuron. Here, modulation depth is defined as:

$$\text{Modulation Depth} = \frac{r_{max} - r_{min}}{\bar{r}} \qquad (2)$$

where

$r_{max}$, $r_{min}$ and $\bar{r}$ are the maximal, minimal and mean firing rate of the tuning curve, respectively.

The stability of tuning was quantified as the correlation coefficient ($r_{xy}$) between the tuning curves computed for each half of the recording session.

**Permutation tests.** The statistical significance of differences between pairs of distributions was assessed using permutation tests. The probability that two observed distributions (y1 and y2) were pulled out of one common distribution was tested as follows. Two vectors (y1 and y2) were randomly selected out of the common vector [y1,y2]. The difference between the two vectors was computed in relation to a pivot point (x0) along the x-axis - either x0 = 0, in cases where x values spanned zero, or x0 = [max(x)-min(x)]/2 otherwise. Then, a difference DD = mean(y1(x < x0)-y2(x < x0)) - mean(y1(x > x0)-y2(x > x0)) was computed for the experimental data and for each of the permutations. The probability of the observed DD to be lower (or higher) than all values obtained in 1000 permutations was calculated and reported as the relevant p-value. The statistical significance of an induced change in the distribution of a variable was tested the same, except that the permutations were done on the original distribution vector rather than on the common vector.

**Cross-correlation and stimulus-triggered average analyses.** Cross-correlations were computed between various simultaneously recorded neuronal (firing rate functions) and kinematic variables using Matlab's xcorr function. Statistical significance was assessed by comparing the raw correlations to those computed from shuffled data. Shuffling of self-generated behavioral variables was done as described in the "Neuronal Tuning Curves to Head and Whisker Kinematics" section. Shuffling of laser stimulation times was done by randomly selecting train onset time within the period of recording that was analyzed. Unless mentioned otherwise, 100 shuffled repetitions were used to estimate chance correlations.

Stimulus-triggered averages (STAs) of neuronal and kinematic variables were computed as follows. The first pulse in each 2 s laser stimulus train (2, 5, or 8 Hz) was considered as the trigger ($t = 0$). Sections of the neuronal or kinematic variable around each occurrence of the trigger (−2–3.8 s) were aligned (neuronal variables were binned according to video frames) and averaged. Standard errors of the mean were calculated across the single-train samples. For each STA, a Modulation Rate (MR) was defined as the mean modulation during stimulation and calculated as MR = mean(abs(diff(STA(0:2 s)))), where the STA was computed as described above.

**Reporting summary**
Further information on research design is available in the Nature Portfolio Reporting Summary linked to this article.

## Data availability
All data supporting the findings of this study are available within this published article and its Supplementary files. Source data are provided with this paper.

## Code availability
Custom codes are available on request from the corresponding authors and in https://doi.org/10.5281/zenodo.12508611.

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

## Acknowledgements

We thank Daniel O'Connor and Alexander Groh for commenting on earlier versions of the manuscript and Arseny Finkelstein and Nachum Ulanovsky for advising on analyzing head kinematics. This research was supported by the United States-Israel Binational Science Foundation (BSF, grant No. 2017216), the Israel Science Foundation (grant No. 2237/20), the Weizmann-UK collaboration grant, and the Yotam project and the Minerva Foundation funded by the Federal German Ministry for Education and Research. E.A. holds the Helen Diller Family Professorial Chair of Neurobiology.

## Author contributions

T.B.O. conceived the study, conducted the experiments, analyzed the data and wrote the paper. A.T. analyzed the data, I.S.S conducted experiments and wrote the paper, E.A. designed the study, conducted and supervised the experiments, analyzed data and wrote the paper. O.Y. helped with study design and supervised the experiments. T.B.O. and A.T. contributed equally to this study.

## Competing interests

The authors declare no competing interests.
