## [Peer Review File · Nature Communications]

REVIEWER COMMENTS

Reviewer #2 (Remarks to the Author):

It does not appear that the authors have substantially or sufficiently dealt with the comments I provided in my first review. In particular, my first major concern regarding the data presented in Figure 4 and the interpretations thereof, largely remains. For publication in a high profile journal like Nature Communications, it would seem necessary to provide a stronger case for the causal involvement of the thalamic nuclei in question.

The text the authors have added, suggesting that the behavioral effects could result from a separate process, is not what I was getting at. The issue is not that the firing induced in thalamic neurons by stimulation is not causally related to changes in head movement; the issue is whether this causal relationship is meaningfully involved during natural behavior of the animal. Meaningful involvement would be shown by optogenetically inactivating the thalamic region and showing a short latency effect on movement. If the activity induced by ChR2 stimulation somehow activates motor pathways or otherwise leads to a startle response that, for example, causes the animal to flinch, there still is a causal relationship between the firing induced and the behavioral effect, but not one that is particularly informative.

In addition, the analysis of time lags in peak correlations, which forms the foundation for the authors existing interpretation, is not convincing for the following reasons:

- If stimulation was indeed causing head movements, this should be clearly apparent as changes in head movement parameters at a short latency following stimulation. But no plots of this kind are provided. The only plot provided is at a low time resolution, so the effect latency cannot be assessed.

- Looking for signs of direct causal relationships using the shape of cross-correlation functions is rather indirect. The authors cite an effect latency with reference to the peak in such a plot, but the correlation values are not much lower at surrounding latencies. The correlation at 0 lag looks to be less than 10% different, and the values at nearby acausal lags, or causal lags that are long and non-specific, look to only be about 20% different. Moreover, the function is not directly compared to a control curve, and no statistics are provided.

- The authors also support their claim using reference to the fraction of cells whose spiking has a peak correlation with the head movement parameters at causal lags, citing increases from values near 30% to values near 50%. There are two problems here. First, no statistics are provided to indicate if these changes are significant. But much more importantly, under a null hypothesis of no causal relationship, the value should approach 50%, since spiking would be unrelated to behavior, and so correlation peaks should be on either side of 0 with equal likelihood. Because we are adding a lot of new spikes here via the stimulation, the relevant comparison is not necessarily the fraction obtained without stimulation. If the new, Chr2-induced spikes are dominating cell firing but induced spiking is unrelated to behavior, we expect 50% causal peaks. This could be addressed with proper statistics, but none are provided.

- The authors refer to changes in correlation peak latency in general and refer to Extended Data Fig. 7, but this figure shows essentially no change for VPM and only shows changes for POM. Moreover, the correlation coefficients peak on either side of 0 for the two parameters shown in relation to POM spiking. Why would a stronger correlation be seen upon stim in the acausal direction if the effect was causal? This is not addressed.

- The language the authors have added to lines 219-225 to clarify their interpretation does not seem sufficiently clear. Are they referring to the causal latencies, or the acausal latencies, that could also reflect expected sensory responses? The inclusion of "and perceiving" here in a section that seems oriented toward motor responses seems to confuse matters.

- It is a bit strange that a unilateral activity perturbation does not cause meaningful change in head angle (a lateralized response) but does cause a change in head linear velocity (a bilateral response). This is not addressed. A bilateral response seems more consistent with a non-specific startle-type response.

- If stimulation effects were specific and causing motor responses, isn't the logical first place to look the whiskers, not the head? It is therefore strange that effects on whiskers are not similarly addressed.

- If we want to make an argument that these thalamic regions are driving head movements, it would also seem logical to look for evidence in the relationship between naturally occurring firing and head (or whisker) movement parameters. However, no such analysis is presented.

Lastly, I would also note that this basic concern about the data presented in Figure 4 was shared in essence by both of the other two reviewers. And again, these concerns do not seem to have been meaningfully addressed.

In addition, my third major concern - "Issues with the language surrounding ascending pathways" - also was not adequately addressed. The language added does not clarify the relevant distinctions between thalamic paralemniscal neurons in POM and the GPR26+ POM focused upon up here. The reference to the paralemniscal pathway in line 253 does not seem justified.

One minor point: on lines 136-7 of the updated manuscript, it is not formally correct to conclude based on the statistical tests for uniformity that "arena exploration did not exhibit directional asymmetry" - rather, what is shown is that it did not exhibit statistically significant directional asymmetry. It is not correct to interpret a p-value above 0.05 as indicating the absence of any sort of structure, or that the null hypothesis is true.

Reviewer #3 (Remarks to the Author):

The revised manuscript of Oram, Tenzer, et al. examine the coding of whisker and head movements in neurons in two thalamic nuclei – the VPM and POM, as well as the behavioral effects of optogenetic stimulation in each region. The manuscript is improved somewhat compared to the original submission although several major concerns have yet to be addressed.

Major comments:

(1) Perhaps the most central results of this study are (from abstract): "in freely moving mice the POM and VPM were involved in whisker kinematic coding to the same degree" and "both thalamic nuclei also robustly coded head kinematics." These results are potentially interesting, but more rigorous analyses are needed to put them on a solid footing.

First, the issue of correlations between kinematic variables persists. I don't quite understand why the authors write that this issue is "not at the focus of this paper, nor is it crucial for the main results presented here." To me, it seems central. The primary conclusion of the study regard the coding of whisker and head kinematics by neurons in the VPM and POM. If one cannot pinpoint which features of movement drive dynamics in VPM and POM, then the impact of the study is greatly diminished. To me, the analytical approach used to address this issue remains confusing. The authors report correlations between kinematic variables that, while small, are highly significant statistically. Then, coding for kinematic variables by single neurons is asserted so long as there is a significant ($p < 0.05$) difference between observed tuning curves and those observed in shuffled data. Using this approach, statistically significant tuning to many kinematic variables would be concluded for many neurons/kinematic variables that are not linked in any real way, but seem so due to the statistics of naturalistic behavior. For examples of general analytical strategies that can be used to address this problem in a rigorous fashion, see, for example, Musall et al (2019) Nat Neuroscience or Campagner et al. (2023) Nature although there are many others.

Related to this issue, I don't understand how the shuffled tuning curves could be observed, if calculated correctly, and it is not clear if this concern affects conclusions of the study (perhaps not, but it is important to know). This is most apparent in Figure 3e/3d since occupancy distributions are reported. For example, if a neuron is observed to have its lowest firing rate when head azimuthal velocity is near zero, and the animal spends most of its time with an azimuthal velocity near zero, then in the shuffled tuning curves, the neuron's firing rate ought to be a little above the minimum firing rate at all azimuthal velocities – not near the maximum firing rate. Perhaps I am missing something, but I still do not understand how the shuffled tuning curves could be near the maximum firing rate in all cases. This comment relates to Figures 2b,c and 3c,e, which articulate the most central conclusions of the study. Also, regarding the construction of shuffled distributions, I understand the authors explanation of Figures 4d,e,f – “the data are not expected to lie within the bootstrapped null distribution in the period “preceding stim” because the stimulations repeated periodically every 2 s (Line 176)” but this an unusual way to present these analyses. More typically, one would determine the bootstrapped null distribution from data recorded in the absence of manipulation to test the null hypothesis that the distribution of a variable of interest is drawn from the ‘baseline’ distribution during periods of stimulation. But issues with Figure 4 are minor compared to those in Figures 2b,c and 3c,e.

The manuscript's presentation of results related to assertions of allocentric coding is not sufficient to support their claims. I realize that this section of the results does not occupy a lot of 'real estate' in the manuscript, but the findings would be very important if substantiated. It would necessitate a re-evaluation of our assumptions about what information is represented in primary sensory nuclei of the thalamus. An allusion is made in Figure 1 to $\theta_{w,a}$, an allocentric whisker variable, and its derivative, but no data are presented regarding tuning to these allocentric whisker variables in Figure 2a-d, and calculation of these variables are not described in the Methods (apologies if I missed it). Figure 2g reports the “difference between its allocentric and whisker-centric MDs, for whisker angle and whisker angular velocity” which I take to mean that the panel labeled ' $\theta_{w,L}$ ' then represents the MD of $\theta_{w,a,L}$ minus the MD of $\theta_{w,L}$. The very surprising finding that neurons are tuned to the angle of whiskers relative to their environment more so than the angle of whiskers relative to the whisker pad would be far better substantiated if additional analyses, if not data, were presented. Minimally, tuning curves need to be included. This finding also seems to imply that at least some VPM/POm neurons would display different activity levels when the whiskers are stationary and at the same angle relative to the pad, but the animal is facing different directions in the environment. Is this interpretation correct? If so, it needs to be demonstrated through additional analyses. See, for example, Figure 2 of Wang, Cheng, and Knierim's 2020 Current Opinion in Neurobiology paper for the kinds of analyses typical for assertions of allocentric coding.

(2) The experiments described in Figure 4 suffer from the same issues noted in my earlier review. In my opinion, these experiments do not add to the manuscript or to our knowledge of thalamic/vibrissal system/motor system function. Similar concerns were raised by all Reviewers and beyond a few added sentences, the fundamental issues with these experiments remain unaddressed. The manuscript makes the argument that because optogenetic stimulation of thalamic neurons sometimes induces movement,

that the VPM and POm are “active players in the sensory-motor loop controlling, and perceiving, head kinematics.” I do not dispute that activation of thalamic nuclei might lead to movement, but this is a trivial and unsurprising result if one is agnostic to the perceptual consequences of this manipulation. For example, if one activated cutaneous receptors in the animal’s tail, one would likely observe an orienting response in animals seeking to determine what it was that just touched them. Yet one would generally not include those receptors in a description of the neural circuits controlling head kinematics. It is the same case here. Sensory information undoubtedly plays an important role in motor control at many levels, but that in itself is not a new finding of this study.

Other comments

1) Figure 1g does not have a legend entry

2) On line 99, head angle does not reflect “compound head and body motion”

3) The sentence beginning “A quantitative comparison of the two populations...” on line 200 could be edited for clarity

4) Line 236-237, “azimuthal tuning of head azimuth” is redundant

5) On line 816, the authors write “a negligible number of short (< 1 ms) inter-spike intervals (< 0.1%).” Presumably the number of short ISIs is reported to provide information about the false positive rate of sorted units. It is important to point out that the number of short ISIs is not the false positive rate. The false positive rate is related to the number of short ISIs, but is orders of magnitude larger (see Hill, Mehta, Kleinfeld), and thus 0.1% of short ISIs is certainly not a ‘negligible’ number. It likely reflects a false positive rate on the order of 10%. This number may be deemed acceptable by the authors, as it is not atypical for unit recordings, but is likely large enough to make cells appear tuned to variables to which they are not in some cases, and this should be made clear.

Point-by-point reply (Nature Communications manuscript NCOMMS-23-42430-T)

Below please find point-by-point replies to all Reviewers' comments.

Reviewers' comments are in **bold**, our replies added as regular text, starting with "Reply:".

Line numbers refer to the file "TessPaper.NatComm.Rev.text.tracked_changes.04" (in "Simple Markup" mode).

We also attach the entire revised passages in each relevant reply.

REVIEWER COMMENTS

Reviewer #2 (Remarks to the Author):

It does not appear that the authors have substantially or sufficiently dealt with the comments I provided in my first review. In particular, my first major concern regarding the data presented in Figure 4 and the interpretations thereof, largely remains. For publication in a high profile journal like Nature Communications, it would seem necessary to provide a stronger case for the causal involvement of the thalamic nuclei in question.

Reply: At the outset we would like to state the following. First, we strongly believe that the paper contains sufficient novel material to justify publication in a high profile journal like Nature Communication. Second, we think that no single publication can cover all controls and tests required for proving causal involvement of neuronal groups in the behavior of freely behaving animals. We certainly agree that our paper only suggests a causal involvement and further studies are required here.

To address the Reviewer's concern in the current paper, we provide additional analysis of the optogenetic stimulation results, as detailed below, based on our existing experimental data. We would like to emphasize that we cannot run any new experiments at this point, given the current national situation in Israel.

The text the authors have added, suggesting that the behavioral effects could result from a separate process, is not what I was getting at. The issue is not that the firing induced in thalamic neurons by stimulation is not causally related to changes in head movement; the issue is whether this causal relationship is meaningfully involved during natural behavior of the animal. Meaningful involvement would be shown by optogenetically inactivating the thalamic region and showing a short latency effect on movement. If the activity induced by ChR2 stimulation somehow activates motor pathways or otherwise leads to a startle response that, for example, causes the animal to flinch, there still is a causal relationship between the firing induced and the behavioral effect, but not one that is particularly informative.

Reply: Three types of startle responses (SRs) are known in rodents: auditory, vestibular and tactile. Auditory and vestibular SRs are ruled out in our experiments. The neuronal pathways conveying tactile SRs are well known - they are conveyed by sensorimotor pathways that pass through the pr5 and sp5 nuclei at the brainstem (García-Hernández & Rubio, 2022; Schmid, Simons, & Schnitzler, 2003; Yeomans, Li, Scott, & Frankland, 2002). VPM and POm do not project back to these brainstem nuclei. Thus, behavioral responses induced by VPM or POm stimulations cannot be considered startle responses because they do not activate the neuronal pathways underlying startle responses. In addition, our control experiments rule out visual startle-like responses.

We modified the text to address this point (Lines 198-200): “Head kinematic changes induced by thalamic stimulations cannot be considered startle responses, as tactile startle responses are conveyed via brainstem trigeminal nuclei,(García-Hernández & Rubio, 2022; Schmid et al., 2003; Yeomans et al., 2002) which do not receive direct thalamic inputs.”

“Meaningful involvement would be shown by optogenetically inactivating the thalamic region and showing a short latency effect on movement.” – We agree that such an experiment would further support causal relationship during natural behavior. But we do not agree that inactivation provides stronger evidence than activation, when both are conducted during natural behavior. First, as it is not possible to activate the entire thalamic nuclei it is not possible to inhibit them altogether. Second, as the VPM&POm contribution to head motion control is only partial, such inhibitions would not generate causal evidence stronger than our activation stimuli. Given those limitations, we believe that a follow-up experiment, aimed at characterizing and quantifying the contributions of VPM and POm to head motion control, requires a full, well-designed study.

In addition, the analysis of time lags in peak correlations, which forms the foundation for the authors existing interpretation, is not convincing for the following reasons:

- If stimulation was indeed causing head movements, this should be clearly apparent as changes in head movement parameters at a short latency following stimulation. But no plots of this kind are provided. The only plot provided is at a low time resolution, so the effect latency cannot be assessed.

Reply: It is not clear to us why the Reviewer states that no plots of this kind are provided. Figure 4c shows latencies between 40 - 80 ms. Figs. 4k,l show latencies in a similar range. We maintain that these latencies are what one would expect for a pathway connecting the sensory thalamus to head motor control. These latencies should be longer than those characterizing startle or other reflexive responses mediated via the brainstem (~ 30 ms; García-Hernández & Rubio, 2022).

In order to strengthen this point we have added four panels showing the effects of optogenetic stimulations on cross-correlations between single-units and head kinematics (Extended data figure 8). The four examples show peaks at the latency range, with neurons both leading and lagging changes in linear velocity, and variable stimulation effects. A corresponding text was added (Lines 230-233) and legend of Extended data figure 8.

The time resolution is dictated by our video frame rate, which was 25 Hz in these experiments. This rate is above the Nyquist requirement for a system with a bandwidth of ~10 Hz.

- Looking for signs of direct causal relationships using the shape of cross-correlation functions is rather indirect. The authors cite an effect latency with reference to the peak in such a plot, but the correlation values are not much lower at surrounding latencies. The correlation at 0 lag looks to be less than 10% different, and the values at nearby acausal lags, or causal lags that are long and non-specific, look to only be about 20% different. Moreover, the function is not directly compared to a control curve, and no statistics are provided.

Reply: The Reviewer probably misses the correct interpretation of the cross-correlation panel. The surrounding latencies that should be compared to the peak correlation are not those of the surrounding peaks, but rather those of the surrounding troughs. The surrounding peaks reflect

mapping of the 2 Hz stimulation autocorrelation onto the crosscorrelation. The correct comparison shows a difference of about 50% (note that this is taken relative to the baseline of the 0.5 Hz periodic correlation).

We should remind the Reviewer that these data are recorded in actively moving, head-free unrestrained animals, in which signal-to-noise ratios for responses to external stimulations are much lower than those expected in passive modes, such as in head-fixed conditions. Given that, we consider the signal-to-noise ratios shown here as high ratios. Finally, the only point of this panel is to show the latency, which, in our eyes, is presented clearly in the panel. Thus, to the best of our judgement, no shuffled correlation is required for convincing that the latency of the central peak was the mean latency between the firing rate and head linear speed in this experiment.

- The authors also support their claim using reference to the fraction of cells whose spiking has a peak correlation with the head movement parameters at causal lags, citing increases from values near 30% to values near 50%. There are two problems here. First, no statistics are provided to indicate if these changes are significant. But much more importantly, under a null hypothesis of no causal relationship, the value should approach 50%, since spiking would be unrelated to behavior, and so correlation peaks should be on either side of 0 with equal likelihood. Because we are adding a lot of new spikes here via the stimulation, the relevant comparison is not necessarily the fraction obtained without stimulation. If the new, Chr2-induced spikes are dominating cell firing but induced spiking is unrelated to behavior, we expect 50% causal peaks. This could be addressed with proper statistics, but none are provided.

Reply: “First, no statistics are provided to indicate if these changes are significant.” – we do provide statistics (Legend of Figure 4k,l; Lines 486-487 of submitted version): “Free behavior and stimulation distributions were significantly different for V_H ($p = 0.03$) and not for $\dot{\theta}_H$ ($p = 0.36$); permutation tests, see Methods”

“under a null hypothesis of no causal relationship” – we do not agree with the Reviewer here. Our null hypothesis here is that there is a sensory causal relationship here and no motor relationship. We do not start from scratch here – we do know, based on the data presented up to this point in the paper, that thalamic neurons do respond to changes in head kinematics. What we do not know is whether they also affect head kinematics. Thus, the assertion of the Reviewer that “If the new, Chr2-induced spikes are dominating cell firing but induced spiking is unrelated to behavior, we expect 50% causal peaks.” Is wrong, because we do know that before adding new Chr2-induced spikes most of the thalamic spikes lag the changes in head kinematics.

- The authors refer to changes in correlation peak latency in general and refer to Extended Data Fig. 7, but this figure shows essentially no change for VPM and only shows changes for POM. Moreover, the correlation coefficients peak on either side of 0 for the two parameters shown in relation to POM spiking. Why would a stronger correlation be seen upon stim in the acausal direction if the effect was causal? This is not addressed.

Reply: Extended Data Fig. 7 was added in order to demonstrate the spectrum of correlation changes observed in the data. In order to address the concern of the Reviewer we have now added Extended Data Fig. 8, which shows several types of changes in correlations between individual neurons and head kinematics (linear velocity). The new figure shows an emergence of

a correlation peak upon stimulation and strengthening of existing peaks, in both the motor and sensory causal directions. Please note that there is no causal and acausal directions here – both sensory and motor directions are causal within the motor-sensory loop.

- The language the authors have added to lines 219-225 to clarify their interpretation does not seem sufficiently clear. Are they referring to the causal latencies, or the acausal latencies, that could also reflect expected sensory responses? The inclusion of “and perceiving” here in a section that seems oriented toward motor responses seems to confuse matters.

Reply: this is now changed to: “It is possible that the observed shift in relative timing between neuronal activity changes and head kinematic changes reflect two unrelated processes, both induced by the stimulations. Yet, it is important to note that the delays observed here between neuronal and kinematic changes (typically between 40 and 120 ms) are longer than those characterizing brainstem-based startle responses³⁴ and shorter than those characterizing cortex-based perceptual reports³⁵. These delays are consistent with the scheme in which the VPM and POm, considered thus far to focus on the processing of sensory vibrissal information, are active players in the sensory-motor loop, controlling and perceiving head kinematics.” (Lines 228-235)

- It is a bit strange that a unilateral activity perturbation does not cause meaningful change in head angle (a lateralized response) but does cause a change in head linear velocity (a bilateral response). This is not addressed. A bilateral response seems more consistent with a non-specific startle-type response.

Reply: As specified above, we believe that startle responses are ruled out here (see our reply to the second comment above). We agree that we do not have a good explanation why the stronger effect is on linear and not lateral motion. We think that there are several possible explanations, but our data do not allow a discrimination between them and thus we do not discuss them. We have now added the text (Line 226): “For reasons that are not yet clear to us,” before “the effect was most evident with V_H , ...”.

- If stimulation effects were specific and causing motor responses, isn't the logical first place to look the whiskers, not the head? It is therefore strange that effects on whiskers are not similarly addressed.

Reply: Our experimental design did not allow addressing motor whisker responses. The experiments of the open field were conducted with long video recordings at 25 Hz. For whisker tracking the frame rate should be > 200 Hz.

- If we want to make an argument that these thalamic regions are driving head movements, it would also seem logical to look for evidence in the relationship between naturally occurring firing and head (or whisker) movement parameters. However, no such analysis is presented.

Reply: The Reviewer is not correct here. Fig. 4h-l panels all include “behavior” and “stimulation” data. “Behavior” refers to data during natural behavior.

Lastly, I would also note that this basic concern about the data presented in Figure 4 was shared in essence by both of the other two reviewers. And again, these concerns do not seem to have been meaningfully addressed.

Reply: indeed all Reviewers refer to Figure 4. However, Reviewer 1 only asked to add a statistical test for Fig. 4l. Reviewer 3 was also mainly concerned about statistical issues related to panels (4d,e,f,j,l). We believe that we have addressed all these concerns appropriately. Reviewer 3 is now also concerned by the appropriate interpretation of the results presented in Fig. 4 – please see our reply to these concerns below.

In addition, my third major concern - “Issues with the language surrounding ascending pathways” - also was not adequately addressed. The language added does not clarify the relevant distinctions between thalamic paralemniscal neurons in POM and the GPR26+ POM focused upon up here. The reference to the paralemniscal pathway in line 253 does not seem justified.

Reply: we have removed the references to the lemniscal and paralemniscal pathways in this passage, which now reads (Lines 260-263): “More specifically, our analysis suggests this processing is biased towards whisker-centric coordinates in the POM and allocentric coordinates in the VPM, which is consistent with the VPM specializing in the processing of external objects and POM in the processing and control of self-motion^{3,37}”

One minor point: on lines 136-7 of the updated manuscript, it is not formally correct to conclude based on the statistical tests for uniformity that “arena exploration did not exhibit directional asymmetry” - rather, what is shown is that it did not exhibit *statistically significant* directional asymmetry. It is not correct to interpret a p-value above 0.05 as indicating the absence of any sort of structure, or that the null hypothesis is true.

Reply: We agree. We have modified the sentence as suggested by the Reviewer.

Reviewer #3 (Remarks to the Author):

The revised manuscript of Oram, Tenzer, et al. examine the coding of whisker and head movements in neurons in two thalamic nuclei – the VPM and POM, as well as the behavioral effects of optogenetic stimulation in each region. The manuscript is improved somewhat compared to the original submission although several major concerns have yet to be addressed.

Major comments:

(1) Perhaps the most central results of this study are (from abstract): “in freely moving mice the POM and VPM were involved in whisker kinematic coding to the same degree” and “both thalamic nuclei also robustly coded head kinematics.” These results are potentially interesting, but more rigorous analyses are needed to put them on a solid footing.

First, the issue of correlations between kinematic variables persists. I don’t quite understand why the authors write that this issue is “not at the focus of this paper, nor is it crucial for the main results presented here.” To me, it seems central. The primary conclusion of the study regard the coding of whisker and head kinematics by neurons in the VPM and POM. If one cannot pinpoint which features of movement drive dynamics in VPM and POM, then the impact of the study is greatly diminished. To me, the analytical approach used to address this issue remains confusing. The authors report correlations between kinematic variables that,

while small, are highly significant statistically. Then, coding for kinematic variables by single neurons is asserted so long as there is a significant ($p < 0.05$) difference between observed tuning curves and those observed in shuffled data. Using this approach, statistically significant tuning to many kinematic variables would be concluded for many neurons/kinematic variables that are not linked in any real way, but seem so due to the statistics of naturalistic behavior. For examples of general analytical strategies that can be used to address this problem in a rigorous fashion, see, for example, Musall et al (2019) Nat Neuroscience or Campagner et al. (2023) Nature although there are many others.

Reply: We do not think that the assertion that “If one cannot pinpoint which features of movement drive dynamics in VPM and POm, then the impact of the study is greatly diminished.” is relevant to the current study, for two reasons. First, the major novelty here, as nicely put by the Reviewer in their opening paragraph, is that VPM and POm code head kinematics. This was not known before and the novel observation necessitates a paradigmatic shift in modeling the functions of these nuclei in tactile perception in the freely moving animal. The decomposition of this coding to its ingredients if of course important, but can be done in follow-up studies. Second, and not unrelated, unlike in head-fixed animals, decomposing the different dynamic features is a very complicated task, requiring a sophisticated design of head-free experiments accompanied with corresponding head-fixed experiments.

“statistically significant tuning to many kinematic variables would be concluded for many neurons/kinematic variables that are not linked in any real way” – again here, we agree but we do not see how this possibility reduces the importance of our main results, as specified by the Reviewer. To be explicit, we have added a note expressing the reservation made by the Reviewer, citing the two mentioned references, in Lines 86-87 (“Note that this criterion does not imply a causal relationship between neuronal activity and kinematic changes^{16,17}”)

Related to this issue, I don't understand how the shuffled tuning curves could be observed, if calculated correctly, and it is not clear if this concern affects conclusions of the study (perhaps not, but it is important to know). This is most apparent in Figure 3e/3d since occupancy distributions are reported. For example, if a neuron is observed to have its lowest firing rate when head azimuthal velocity is near zero, and the animal spends most of its time with an azimuthal velocity near zero, then in the shuffled tuning curves, the neuron's firing rate ought to be a little above the minimum firing rate at all azimuthal velocities – not near the maximum firing rate. Perhaps I am missing something, but I still do not understand how the shuffled tuning curves could be near the maximum firing rate in all cases. This comment relates to Figures 2b,c and 3c,e, which articulate the most central conclusions of the study. Also, regarding the construction of shuffled distributions, I understand the authors explanation of Figures 4d,e,f – “the data are not expected to lie within the bootstrapped null distribution in the period “preceding stim” because the stimulations repeated periodically every 2 s (Line 176)” but this an unusual way to present these analyses. More typically, one would determine the bootstrapped null distribution from data recorded in the absence of manipulation to test the null hypothesis that the distribution of a variable of interest is drawn from the ‘baseline’ distribution during periods of stimulation. But issues with Figure 4 are minor compared to those in Figures 2b,c and 3c,e.

Reply: Right, we indeed were not clear enough about the scaling for the shuffled curves. Thank you for this comment. In response to a previous revision request, we transitioned from using

normalized values ([0, 1]) to real values but failed to provide a corresponding clarification for the shuffled curves. This oversight has been now rectified as follows: in the legend of the respective panels (2b and 3c, e), we have included the following statement: “Each curve is normalized to its maximal value. The firing rates on the ordinate in each plot pertain exclusively to the empirical data.” We believe that this addition addresses the Reviewer's concerns and clarifies the scaling of the shuffled curves.”

We also thank the reviewer for proposing an alternative (likely better) way to the construct the shuffled distributions in Fig. 4d,e,f. Given the minor difference in this case we chose to leave the current construction.

The manuscript’s presentation of results related to assertions of allocentric coding is not sufficient to support their claims. I realize that this section of the results does not occupy a lot of ‘real estate’ in the manuscript, but the findings would be very important if substantiated. It would necessitate a re-evaluation of our assumptions about what information is represented in primary sensory nuclei of the thalamus. An allusion is made in Figure 1 to $\theta_{w,a}$, an allocentric whisker variable, and its derivative, but no data are presented regarding tuning to these allocentric whisker variables in Figure 2a-d, and calculation of these variables are not described in the Methods (apologies if I missed it). Figure 2g reports the “difference between its allocentric and whisker-centric MDs, for whisker angle and whisker angular velocity” which I take to mean that the panel labeled ‘ $\theta_{w,L}$ ’ then represents the MD of $\theta_{w,a,L}$ minus the MD of $\theta_{w,L}$. The very surprising finding that neurons are tuned to the angle of whiskers relative to their environment more so than the angle of whiskers relative to the whisker pad would be far better substantiated if additional analyses, if not data, were presented. Minimally, tuning curves need to be included. This finding also seems to imply that at least some VPM/POM neurons would display different activity levels when the whiskers are stationary and at the same angle relative to the pad, but the animal is facing different directions in the environment. Is this interpretation correct? If so, it needs to be demonstrated through additional analyses. See, for example, Figure 2 of Wang, Cheng, and Knierim’s 2020 Current Opinion in Neurobiology paper for the kinds of analyses typical for assertions of allocentric coding.

Reply: We agree with the Reviewer that “the findings would be very important if substantiated”. We should emphasize, however, that such a substantiation must be done in a follow-up study. There is a limit to what can be covered in a single study/publication and we have reached this limit for the current study. We believe that there are enough novel findings in the current manuscript, and enough background for guiding such follow-up studies.

“calculation of these variables are not described in the Methods (apologies if I missed it)” – We thank the Reviewer for this comment and apologize for our oversight. The description is now added (Lines 886-889): “Two allocentric kinematic variables were analyzed for the median angle of the whisker beam on each side of the snout: whisker azimuth in allocentric coordinates ($\theta_{w,a}$) and allocentric whisker azimuthal velocity ($\dot{\theta}_{w,a}$). Allocentric whisker variables were calculated by summing the concurrent corresponding head and whisker variables.” A reference to this text, as well as explicit explanation using the Reviewer suggested phrasing, was added in the legend of Fig. 2f.

“at least some VPM/POM neurons would display different activity levels when the whiskers are stationary and at the same angle relative to the pad, but the animal is facing different directions

in the environment” – Right. This is indeed expected for many of the units showing significant and stable coding of head azimuth (e.g., Fig. 3c,e). We have added a citation of the Wang paper in this respect (Line 121).

“Minimally, tuning curves need to be included.” We calculated allocentric tuning curves for the four variables described here. We have now added an Extended data figure (**Extended data figure 9**) that shows 16 such curves, 8 per nucleus and 4 per variable. We have modified the text accordingly (Lines 122-125).

(2) The experiments described in Figure 4 suffer from the same issues noted in my earlier review. In my opinion, these experiments do not add to the manuscript or to our knowledge of thalamic/vibrissal system/motor system function. Similar concerns were raised by all Reviewers and beyond a few added sentences, the fundamental issues with these experiments remain unaddressed. The manuscript makes the argument that because optogenetic stimulation of thalamic neurons sometimes induces movement, that the VPM and POm are “active players in the sensory-motor loop controlling, and perceiving, head kinematics.” I do not dispute that activation of thalamic nuclei might lead to movement, but this is a trivial and unsurprising result if one is agnostic to the perceptual consequences of this manipulation. For example, if one activated cutaneous receptors in the animal’s tail, one would likely observe an orienting response in animals seeking to determine what it was that just touched them. Yet one would generally not include those receptors in a description of the neural circuits controlling head kinematics. It is the same case here. Sensory information undoubtedly plays an important role in motor control at many levels, but that in itself is a not a new finding of this study.

Reply: The concerns of the Reviewers about the data presented in Figure 4 varied. Reviewer 1 asked to add statistical tests for the results presented in panel 4l. Reviewer 2 was concerned about the possibility that motor responses were conveyed by alternative motor pathways, such as those conveying startle responses. Reviewer 3 was previously concerned by statistical issues related to panels 4d,e,f,j,l. Reviewer 3 now emphasizes the concern, which we have likely missed before, about the possibility that the observed motor responses are of a higher-order kind, involving the level of perceptual interpretation of the sensory stimuli. All concerns are in place and we thank the Reviewers for bringing them up.

We have addressed the previous concern of Reviewer 3 as well as the concerns of the other two Reviewers above and in our previous revision. As for the current concern of Reviewer 3 we claim that the latencies observed here rule out the possibility raised by the Reviewer. Perceptual decisions in mice, presumably involving significant cortical processing, are typically characterized by delays > 150 ms (Neubarth et al.). The delays in our study were typically < 120 ms (Fig. 4c,j,k,l; Extended Data Figure 8). To clarify this point we have modified the passage in Lines 230-233 to read: “Yet, it is important to note that the delays observed here between neuronal and kinematic changes (typically between 40 and 120 ms) are longer than those characterizing brainstem-based startle responses³² and shorter than those characterizing cortex-based perceptual reports³³.”

Other comments

1) Figure 1g does not have a legend entry

Reply: fixed

2) On line 99, head angle does not reflect “compound head and body motion”

Reply : we measure the allocentric head angle (over time), which does reflect the compound head and body motion.

3) The sentence beginning “A quantitative comparison of the two populations...” on line 200 could be edited for clarity

Reply: fixed

4) Line 236-237, “azimuthal tuning of head azimuth” is redundant

Reply: fixed

5) On line 816, the authors write “a negligible number of short (< 1 ms) inter-spike intervals (< 0.1%).” Presumably the number of short ISIs is reported to provide information about the false positive rate of sorted units. It is important to point out that the number of short ISIs is not the false positive rate. The false positive rate is related to the number of short ISIs, but is orders of magnitude larger (see Hill, Mehta, Kleinfeld), and thus 0.1% of short ISIs is certainly not a ‘negligible’ number. It likely reflects a false positive rate on the order of 10%. This number may be deemed acceptable by the authors, as it is not atypical for unit recordings, but is likely large enough to make cells appear tuned to variables to which they are not in some cases, and this should be made clear.

Reply: A clarifying note was added (“; this value should not be confused with the false positive rate of spike detection”).

References

- García-Hernández, S., & Rubio, M. E. (2022). Role of GluA4 in the acoustic and tactile startle responses. *Hearing Research*, *414*, 108410.
- Neubarth, N. L., Emanuel, A. J., Liu, Y., Springel, M. W., Handler, A., Zhang, Q., . . . Ginty, D. D. (2020). Meissner corpuscles and their spatially intermingled afferents underlie gentle touch perception. *Science*, *368*(6497).
- Schmid, S., Simons, N. S., & Schnitzler, H. U. (2003). Cellular mechanisms of the trigeminally evoked startle response. *European Journal of Neuroscience*, *17*(7), 1438-1444.
- Yeomans, J. S., Li, L., Scott, B. W., & Frankland, P. W. (2002). Tactile, acoustic and vestibular systems sum to elicit the startle reflex. *Neuroscience & Biobehavioral Reviews*, *26*(1), 1-11.

REVIEWER COMMENTS

Reviewer #2 (Remarks to the Author):

Substantial issues remain with the interpretation of the optogenetic data and the statistical foundation for the claims that spiking is causally related to head movement. Below I respond to the authors replies.

Reviewer #2 (Remarks to the Author):

It does not appear that the authors have substantially or sufficiently dealt with the comments I provided in my first review. In particular, my first major concern regarding the data presented in Figure 4 and the interpretations thereof, largely remains. For publication in a high profile journal like Nature Communications, it would seem necessary to provide a stronger case for the causal involvement of the thalamic nuclei in question.

Reply: At the outset we would like to state the following. First, we strongly believe that the paper contains sufficient novel material to justify publication in a high profile journal like Nature Communication. Second, we think that no single publication can cover all controls and tests required for proving causal involvement of neuronal groups in the behavior of freely behaving animals. We certainly agree that our paper only suggests a causal involvement and further studies are required here.

To address the Reviewer's concern in the current paper, we provide additional analysis of the optogenetic stimulation results, as detailed below, based on our existing experimental data. We would like to emphasize that we cannot run any new experiments at this point, given the current national situation in Israel.

My assessment is that the paper as currently composed does not provide strong enough support for the causal involvement of the pathways in question in driving motor responses during behavior. It is broadly accepted, at least in the field of motor control, that stimulation, whether electrical, optogenetic, or otherwise, can induce activity states that do not occur naturally, and that stimulation effects can reflect pathway activation that does not occur naturally to any substantial degree.

The text the authors have added, suggesting that the behavioral effects could result from a separate process, is not what I was getting at. The issue is not that the firing induced in thalamic neurons by stimulation is not causally related to changes in head movement; the issue is whether this causal relationship is meaningfully involved during natural behavior of the animal. Meaningful involvement would be shown by optogenetically inactivating the thalamic region and showing a short latency effect on movement. If the activity induced by ChR2 stimulation somehow activates motor pathways or otherwise leads to a startle response that, for example, causes the animal to flinch, there still is a causal relationship between the firing induced and the behavioral effect, but not one that is particularly informative.

Reply: Three types of startle responses (SRs) are known in rodents: auditory, vestibular and tactile. Auditory and vestibular SRs are ruled out in our experiments. The neuronal pathways conveying tactile SRs are well known - they are conveyed by sensorimotor pathways that pass through the pr5 and sp5 nuclei at the brainstem (García-Hernández & Rubio, 2022; Schmid, Simons, & Schnitzler, 2003; Yeomans, Li, Scott, & Frankland, 2002). VPM and POr do not project back to these brainstem nuclei. Thus, behavioral responses induced by VPM or POr stimulations cannot be considered startle responses because they do not activate the neuronal pathways underlying startle responses. In addition, our control experiments rule out visual startle-like responses.

We modified the text to address this point (Lines 198-200): “Head kinematic changes induced by thalamic stimulations cannot be considered startle responses, as tactile startle responses are conveyed via brainstem trigeminal nuclei,(García-Hernández & Rubio, 2022; Schmid et al., 2003; Yeomans et al., 2002) which do not receive direct thalamic inputs.”

“Meaningful involvement would be shown by optogenetically inactivating the thalamic region and showing a short latency effect on movement.” – We agree that such an experiment would further support causal relationship during natural behavior. But we do not agree that inactivation provides stronger evidence than activation, when both are conducted during natural behavior. First, as it is not possible to activate the entire thalamic nuclei it is not possible to inhibit them altogether. Second, as the VPM&POr contribution to head motion control is only partial, such inhibitions would not generate causal evidence stronger than our activation stimuli. Given those limitations, we believe that a follow-up experiment, aimed at characterizing and quantifying the contributions of VPM and POr to head motion control, requires a full, well-designed study.

That what we might call ‘startle’ only occurs naturally in response to sensory stimuli does not diminish the concern in this case. Here, the authors are inducing a substantial change of activity suddenly in the brain, which could generate a response in the brain with a behavioral consequence. This behavioral consequence could result from activation of relatively direct efferent pathways leading to brainstem motor circuits and then to muscles, or through less direct pathways. The latter type could include the detection of substantial unexpected neural activity inducing something like a startle. But regardless of

the extent to which the response mechanistically aligns with naturally occurring startle responses, this latter class of so-called 'indirect' effects are a substantial concern, of which much has been written over the last several years (cf. Otchy et al, Nature, 2015).

But it is not just the possibility of non-specific or indirect effects that is relevant here. Even if the stimulation is activating relatively direct pathways to the motor periphery, it does not indicate that such pathways are also activated by thalamus during naturally-occurring behavior.

In addition, the analysis of time lags in peak correlations, which forms the foundation for the authors existing interpretation, is not convincing for the following reasons:

- If stimulation was indeed causing head movements, this should be clearly apparent as changes in head movement parameters at a short latency following stimulation. But no plots of this kind are provided. The only plot provided is at a low time resolution, so the effect latency cannot be assessed.

Reply: It is not clear to us why the Reviewer states that no plots of this kind are provided. Figure 4c shows latencies between 40 - 80 ms. Figs. 4k,l show latencies in a similar range. We maintain that these latencies are what one would expect for a pathway connecting the sensory thalamus to head motor control. These latencies should be longer than those characterizing startle or other reflexive responses mediated via the brainstem (~ 30 ms; García-Hernández & Rubio, 2022).

In order to strengthen this point we have added four panels showing the effects of optogenetic stimulations on cross-correlations between single-units and head kinematics (Extended data figure 8). The four examples show peaks at the latency range, with neurons both leading and lagging changes in linear velocity, and variable stimulation effects. A corresponding text was added (Lines 230-233) and legend of Extended data figure 8.

The time resolution is dictated by our video frame rate, which was 25 Hz in these experiments. This rate is above the Nyquist requirement for a system with a bandwidth of ~10 Hz.

First of all, there is no labeling of the time base in the inset of Figure 4c, so it is not clear where 40-80 ms is coming from. Second, according to the legend for 4c, it uses the data for the example presented in 4a. If it was clear where the 40-80 ms range was coming from, it would still not be apparent how reproducible this range is across replicates, nor the basis for the authors' statement that "neuronal changes typically led head velocity changes". Lastly, to accept that the causal lag here is meaningful, it would need to reflect a significant change from the correlation in the absence of stimulation, but that correlation is not shown.

The data in Figure 4k and l also do not unambiguously support the conclusion that stimulation specifically causes a change at 40-80 ms. The statistical test done to show a significant change between the control and stimulation distribution tests the entire distribution, not the values at 40 or 80 ms specifically – as far as I can tell from the Methods. Therefore, the differences could result, at least in part, from differences at other latencies. The distributions shown in 4k and l lack error bars, so the actual differences between conditions not due to chance are not clear. Perhaps the authors could do a test just on the numbers of cells that show peaks in a certain range or at a certain value, or treat each animal as a replicate and test the difference across animals. The examples added in ED Figure 8 help illustrate the measurements, but as examples, they do not address the significance of the changes.

In general, the expected latency depends on the pathways mediating the effects. If the latency of the effect here cannot be distinguished from the latency of nonspecific or indirect effects, this challenges the interpretation that stimulation is revealing pathways that directly link whisker thalamus to head movements.

- Looking for signs of direct causal relationships using the shape of cross-correlation functions is rather indirect. The authors cite an effect latency with reference to the peak in such a plot, but the correlation values are not much lower at surrounding latencies. The correlation at 0 lag looks to be less than 10% different, and the values at nearby acausal lags, or causal lags that are long and non-specific, look to only be about 20% different. Moreover, the function is not directly compared to a control curve, and no statistics are provided.

Reply: The Reviewer probably misses the correct interpretation of the cross-correlation panel. The surrounding latencies that should be compared to the peak correlation are not those of the surrounding peaks, but rather those of the surrounding troughs. The surrounding peaks reflect mapping of the 2 Hz stimulation autocorrelation onto the crosscorrelation. The correct comparison shows a difference of about 50% (note that this is taken relative to the baseline of the 0.5 Hz periodic correlation).

We should remind the Reviewer that these data are recorded in actively moving, head-free unrestrained animals, in which signal-to-noise ratios for responses to external stimulations are much lower than those expected in passive modes, such as in head-fixed conditions. Given that, we consider the signal-to-noise ratios shown here as high ratios. Finally, the only point of this panel is to show the latency, which, in our eyes, is presented clearly in the panel. Thus, to the best of our judgement, no shuffled correlation is required for convincing that the latency of the central peak was the mean latency between the firing rate and head linear speed in this experiment.

The authors have not understood my concern. I was not referring to the surrounding peaks, but to surrounding points. The inset in 4c shows that the curve is very nearly symmetric around zero. That the peak falls slightly to the right of 0 using the data from the example shown in panel a (this is according to the legend) is not statistically founded evidence that spiking in the thalamic cells leads head movement at a short causal lag. Replicating this measurement across animals (the authors refer to measuring the effects “in several cases”) would provide a statistical foundation for the statement about the 40-80 ms causal lag in the Results. This should not require collecting more data.

- The authors also support their claim using reference to the fraction of cells whose spiking has a peak correlation with the head movement parameters at causal lags, citing increases from values near 30% to values near 50%. There are two problems here. First, no statistics are provided to indicate if these changes are significant. But much more importantly, under a null hypothesis of no causal relationship, the value should approach 50%, since spiking would be unrelated to behavior, and so correlation peaks should be on either side of 0 with equal likelihood. Because we are adding a lot of new spikes here via the stimulation, the relevant comparison is not necessarily the fraction obtained without stimulation. If the new, ChR2-induced spikes are dominating cell firing but induced spiking is unrelated to behavior, we expect 50% causal peaks. This could be addressed with proper statistics, but none are provided.

Reply: “First, no statistics are provided to indicate if these changes are significant.” – we do provide statistics (Legend of Figure 4k,l; Lines 486-487 of submitted version): “Free behavior and stimulation distributions were significantly different for VH ($p = 0.03$) and not for θ_H ($p = 0.36$); permutation tests, see Methods”

I thank the author for pointing this out – I had missed the statistical tests quoted in the legend. However, as I describe above, because of the test used, it does not specifically indicate a significant change in neurons with firing rates leading head movement.

“under a null hypothesis of no causal relationship” – we do not agree with the Reviewer here. Our null hypothesis here is that there is a sensory causal relationship here and no motor relationship. We do not start from scratch here – we do know, based on the data presented up to this point in the paper, that thalamic neurons do respond to changes in head kinematics. What we do not know is whether they also affect head kinematics. Thus, the assertion of the Reviewer that “If the new, ChR2-induced spikes are dominating cell firing but induced spiking is unrelated to behavior, we expect 50% causal peaks.” Is wrong, because we do know that before adding new ChR2-induced spikes most of the thalamic spikes lag the changes in head kinematics.

Respectfully, my assertion quoted at the end there is not wrong. The spikes induced by ChR2 stimulation are not caused by sensation, so there are two possibilities for those spikes – they can cause movement, or they can be unrelated to movement. If the former, we expect a causal lag between spike times and head movement. If the latter, we expect the spikes to be equally likely at causal and acausal lags. Note that here by “acausal” I mean from the perspective of the neuron – you could replace these terms with “motor” and “sensory” if you prefer. Now, imagine a scenario in which during normal behavior, firing is driven by sensation, and so spiking lags head movements on average. If we induce spiking with ChR2 in very large amounts relative to the firing during behavior so that the vast majority of spikes are induced by ChR2, what do we expect under our two scenarios? If the induced spiking is driving head movements, we expect spiking mostly at a causal lag. If the induced spiking and head movements are not causally related, we expect an equal number of causal and acausal lags.

Therefore, in order for a change in the number of causal peaks from 30% to 50% to be evidence of causal influence, we need to know how much of the spikes in these calculations are induced, versus naturally occurring. The authors should be able to estimate this to provide evidence that the induced spikes are primarily at causal lags.

That the induced spikes do not dominate does not look to be easily assumed. The example shown in 4a shows that the firing rates are much higher during stimulation. The pulses are brief, which might mean that induced spikes do not meaningfully dominate here, but the authors would need to show that.

- The authors refer to changes in correlation peak latency in general and refer to Extended Data Fig. 7, but this figure shows essentially no change for VPM and only shows changes for P0m. Moreover, the correlation coefficients peak on either side of 0 for the two parameters shown in relation to P0m spiking. Why would a stronger correlation be seen upon stim in the acausal direction if the effect was causal? This is not addressed.

Reply: Extended Data Fig. 7 was added in order to demonstrate the spectrum of correlation changes observed in the data. In order to address the concern of the Reviewer we have now added Extended Data Fig. 8, which shows several types of changes in correlations between individual neurons and head kinematics (linear velocity). The new figure shows an emergence of a correlation peak upon stimulation and strengthening of existing peaks, in both the motor and sensory causal directions. Please note that there is no causal and acausal directions here – both sensory and motor directions are causal within the motor-sensory loop.

The authors have not addressed the concerns that the correlations do not seem to change for VPM upon ChR2 stimulation (Extended Data Figure 7a,c) in the range (< 200 ms) stated in the text that references this figure (lines 215-218).

It is also not clear exactly what structure in Extended Data Figure 7b,d supports the authors claim either. Panel b does not seem to show an increase in correlation biased in the causal direction, and so seems equally consistent with an absence of causal effects on head movements. If the only support for the authors claim comes from panel d, the text should make that clear, and the relevant structure in panel d that supports that should also be clarified. Since the chance distributions for the Behavior and Stimulation data are so different (likely because much more Behavior data is used), it is hard to assess the real differences in the two curves plotted.

- The language the authors have added to lines 219-225 to clarify their interpretation does not seem sufficiently clear. Are they referring to the causal latencies, or the acausal latencies, that could also reflect expected sensory responses? The inclusion of "and perceiving" here in a section that seems oriented toward motor responses seems to confuse matters.

Reply: this is now changed to: "It is possible that the observed shift in relative timing between neuronal activity changes and head kinematic changes reflect two unrelated processes, both induced by the stimulations. Yet, it is important to note that the delays observed here between neuronal and kinematic changes (typically between 40 and 120 ms) are longer than those characterizing brainstem-based startle responses³⁴ and shorter than those characterizing cortex-based perceptual reports³⁵. These delays are consistent with the scheme in which the VPM and POm, considered thus far to focus on the processing of sensory vibrissal information, are active players in the sensory-motor loop, controlling and perceiving head kinematics." (Lines 228-235)

As I describe above, distinguishing from the latencies of naturally occurring startle is not sufficient to relieve my concern. I would also note that natural startles would involve sensory reception and peripheral conduction, which would not be part of the latency here. The larger problem here though is that a statistical foundation for the observation of increased causal correlation is not sufficiently strong. I do believe it may be possible to present a sound statistical argument with the existing data however.

- If we want to make an argument that these thalamic regions are driving head movements, it would also seem logical to look for evidence in the relationship between naturally occurring firing and head (or whisker) movement parameters. However, no such analysis is presented.

Reply: The Reviewer is not correct here. Fig. 4h-l panels all include “behavior” and “stimulation” data. “Behavior” refers to data during natural behavior.

My general point is that it would be helpful to show that naturally occurring activity is consistent with causal influence on head movement, and not just implicitly. The author is technically correct in their reply here, but these data are all presented in the context of the optogenetic results. 4h is at low time resolution, so the Behavior curve is hard to read at the relevant lags. 4i is unrelated to neural activity. 4j combines data from both free behavior and stimulation, such that it is not clear that causal lags are present during free behavior. The one panel that does include cells showing causal lags during free behavior is described only with the observation that most neurons showed acausal/sensory lags, as a counterpoint to the results in the next panel.

Reviewer #3 (Remarks to the Author):

The revised manuscript of Oram, Tenzer et al. is modestly improved but the central issues identified by myself and the other reviewers largely remain. The authors’ response mostly presents arguments with critiques of the reviewers in lieu of improvements to the manuscript, and in most cases, I do not find these arguments compelling. The central result regarding coding of head kinematics still suffers from the issue of correlations between kinematic variables, and so it is not clear that the authors have correctly identified that head kinematics are the thing being encoded. Demonstration of allocentric coding of whiskers remains unconvincing. Some results of the study may well be correct, but it is impossible to know with any certainty given the data and analyses presented.

Finally, all three reviewers noted that the results of Figure 4 should not be interpreted in the way the manuscript interprets them. After two rounds of reviews, it remains essentially unchanged. All three reviewers articulated these concerns – albeit in different ways – but all comments were fundamentally focused on the same problem. I very much agree with the statements made by the other reviewers and echoed my own. Because the problem was described in different ways by the different reviewers, the Authors make it seem like different criticisms were levied by different reviewers on different rounds of review, but I do not see it this way. In the latest revision, the authors interpret concerns of myself and R2 overly narrowly so as to refute them, for example, with respect to addressing the startle-like responses mentioned by R2. R2’s point persists if the response to optogenetic stimulation is not a literal startle

response mediated only by medullary circuits, but merely startling to the animal in some way. The authors seem to suggest, in response to my previous review, that the response of a mouse to external stimuli can only fall into the categories of 'startle responses' and 'perceptual decisions' with latencies >150 ms. Mice can certainly respond to external stimuli with learned movements in 50-100 ms and innate movements to strong stimuli are likely faster. A mouse orienting to identify what just touched its tail certainly does not require a perceptual decision in the sense that that term is usually used in the literature. I don't know why 'significant cortical processing' would be assumed.

Rebuttal point-by-point reply to reviewers' comments

Text from previous cycles are in *Italics*. New comments of the Reviewers begin with [Reviewer:]. Our new replies are in **blue text** and start with "**REPLY:**"

Reviewer #2 (Remarks to the Author):

Substantial issues remain with the interpretation of the optogenetic data and the statistical foundation for the claims that spiking is causally related to head movement. Below I respond to the authors replies.

Reviewer #2 (Remarks to the Author):

It does not appear that the authors have substantially or sufficiently dealt with the comments I provided in my first review. In particular, my first major concern regarding the data presented in Figure 4 and the interpretations thereof, largely remains. For publication in a high profile journal like Nature Communications, it would seem necessary to provide a stronger case for the causal involvement of the thalamic nuclei in question.

Reply: At the outset we would like to state the following. First, we strongly believe that the paper contains sufficient novel material to justify publication in a high profile journal like Nature Communication. Second, we think that no single publication can cover all controls and tests required for proving causal involvement of neuronal groups in the behavior of freely behaving animals. We certainly agree that our paper only suggests a causal involvement and further studies are required here.

To address the Reviewer's concern in the current paper, we provide additional analysis of the optogenetic stimulation results, as detailed below, based on our existing experimental data. We would like to emphasize that we cannot run any new experiments at this point, given the current national situation in Israel.

[Reviewer:] My assessment is that the paper as currently composed does not provide strong enough support for the causal involvement of the pathways in question in driving motor responses during behavior. It is broadly accepted, at least in the field of motor control, that stimulation, whether electrical, optogenetic, or otherwise, can induce activity states that do not occur naturally, and that stimulation effects can reflect pathway activation that does not occur naturally to any substantial degree.

REPLY: We accept the reservation of the Reviewer. We have thus added this reservation to the paper. After (Lines 233-236) "These delays are consistent with the possibility that the VPM and POm, considered thus far to focus on the processing of sensory vibrissal information, might be active players in the sensory-motor loop, mediating the perception of head kinematics." We have added (Lines 236-238): "Yet, it is important to note that stimulations of neuronal groups can induce activity states that do not occur naturally and can activate pathways that are not activated during natural behavior."

We of course agree with the general notion that animal behavior in response to unnatural brain stimulations cannot be assumed to be identical, and maybe not even similar, to their responses to natural stimuli. We agree. But isn't this always a concern with perturbation experiments? We take our excitatory perturbations as suggestive, not as a proof, for the involvement of the sensory vibrissal thalamus in affecting the dynamics of head kinematics, and consider our results as a first step in this direction, inviting additional, more detailed, future experiments.

Thanks to the Reviewers' (justified) concerns, we realized that our use of the terminology of "motor control" is not justified and indeed might be misleading. Thus, throughout the paper, we have replaced this terminology with one expressing the potential effect of thalamic activity on head kinematics, better conveying the proposed function of the sensory thalamic nuclei in active perception of head kinematics. We also emphasize now that more definitive answers require further experimentation. The actual changes are (the essential changes are underlined):

1. End of Abstract (Lines 21-24): we changed "**Collectively, our data suggest that the VPM and POm both process ... and participate in the control of head motion.**" to "**Collectively, our data suggest that the VPM and POm both process ... and can affect the dynamics of head kinematics.**" (the addition of "can" here emphasizes that the animals do not necessarily use this ability during natural behavior).
2. Lines 172: we changed "In order to study the involvement of these thalamic neurons in head-motion control loops, loops in which these neurons both process sensory information and affect motor activity, we tracked ..." to "In order to test whether these thalamic neurons can affect head kinematics, we tracked ..."
3. Lines 222-223: we changed "This suggests a scheme in which thalamic neurons of the vibrissal sensory nuclei participate in functional loops that control head kinematics" to "This suggests a scheme in which thalamic neurons of the vibrissal sensory nuclei participate in functional loops that affect head kinematics".
4. Lines 233-236: changed "These delays are consistent with the scheme in which the VPM and POm, considered thus far to focus on the processing of sensory vibrissal information, are active players in the sensory-motor loop, controlling and perceiving head kinematics." To "These delays are consistent with the possibility that the VPM and POm, considered thus far to focus on the processing of sensory vibrissal information, might be active players in the sensory-motor loop, mediating the perception of head kinematics."
5. Line 245: deleted "Not only that, but the activity profile of VPM and POm neurons is consistent with these nuclei participating in sensory-motor loops controlling head motion (the reasonable possibility that they also participate in loops controlling whisking was not tested here)."
6. Line 284-287: changed "further investigations are warranted to unravel the specific dynamical processes involved in object perception within these nuclei and to elucidate the division of labor between them." to "further investigations are warranted to unravel the specific dynamical processes involved in object

perception within these nuclei, to elucidate the division of labor between them and to clarify their potential effects on head and whiskers kinematics.’

The text the authors have added, suggesting that the behavioral effects could result from a separate process, is not what I was getting at. The issue is not that the firing induced in thalamic neurons by stimulation is not causally related to changes in head movement; the issue is whether this causal relationship is meaningfully involved during natural behavior of the animal. Meaningful involvement would be shown by optogenetically inactivating the thalamic region and showing a short latency effect on movement. If the activity induced by ChR2 stimulation somehow activates motor pathways or otherwise leads to a startle response that, for example, causes the animal to flinch, there still is a causal relationship between the firing induced and the behavioral effect, but not one that is particularly informative.

Reply: Three types of startle responses (SRs) are known in rodents: auditory, vestibular and tactile. Auditory and vestibular SRs are ruled out in our experiments. The neuronal pathways conveying tactile SRs are well known - they are conveyed by sensorimotor pathways that pass through the pr5 and sp5 nuclei at the brainstem (García-Hernández & Rubio, 2022; Schmid, Simons, & Schnitzler, 2003; Yeomans, Li, Scott, & Frankland, 2002). VPM and POM do not project back to these brainstem nuclei. Thus, behavioral responses induced by VPM or POM stimulations cannot be considered startle responses because they do not activate the neuronal pathways underlying startle responses. In addition, our control experiments rule out visual startle-like responses.

We modified the text to address this point (Lines 198-200): “Head kinematic changes induced by thalamic stimulations cannot be considered startle responses, as tactile startle responses are conveyed via brainstem trigeminal nuclei, (García-Hernández & Rubio, 2022; Schmid et al., 2003; Yeomans et al., 2002) which do not receive direct thalamic inputs.”

“Meaningful involvement would be shown by optogenetically inactivating the thalamic region and showing a short latency effect on movement.” – We agree that such an experiment would further support causal relationship during natural behavior. But we do not agree that inactivation provides stronger evidence than activation, when both are conducted during natural behavior. First, as it is not possible to activate the entire thalamic nuclei it is not possible to inhibit them altogether. Second, as the VPM&POM contribution to head motion control is only partial, such inhibitions would not generate causal evidence stronger than our activation stimuli. Given those limitations, we believe that a follow-up experiment, aimed at characterizing and quantifying the contributions of VPM and POM to head motion control, requires a full, well-designed study.

[Reviewer:] That what we might call ‘startle’ only occurs naturally in response to sensory stimuli does not diminish the concern in this case. Here, the authors are inducing a substantial change of activity suddenly in the brain, which could generate a response in the brain with a behavioral consequence. This behavioral consequence could result from

activation of relatively direct efferent pathways leading to brainstem motor circuits and then to muscles, or through less direct pathways. The latter type could include the detection of substantial unexpected neural activity inducing something like a startle. But regardless of the extent to which the response mechanistically aligns with naturally occurring startle responses, this latter class of so-called ‘indirect’ effects are a substantial concern, of which much has been written over the last several years (cf. Otchy et al, Nature, 2015).

But it is not just the possibility of non-specific or indirect effects that is relevant here. Even if the stimulation is activating relatively direct pathways to the motor periphery, it does not indicate that such pathways are also activated by thalamus during naturally-occurring behavior.

REPLY: The Reviewer suggests that “This behavioral consequence could result from activation of relatively direct efferent pathways leading to brainstem motor circuits and then to muscles, or through less direct pathways.” As we have stated in our previous reply and in the manuscript, there are no efferent connections between the sensory thalamus and the sensory brainstem (which conveys tactile startle responses). We indeed agree that thalamic neurons can activate brainstem motor circuits – in fact, this is one possible path through which our thalamic neurons could participate in affecting head kinematics.

We thank the Reviewer for mentioning Otchy et al’s paper – this paper indeed clearly supports the reservation raised by the Reviewer – we now cite this paper in Line 238, as a reference for the Reviewer’s proposed reservation.

We also agree with the additional reservation raised here by the Reviewer. We believe that our addition (Lines 236-238) expresses this reservation (“Yet, it is important to note that artificial stimulations of neuronal groups can induce activity states that do not occur naturally and can activate pathways that are not activated during natural behavior³⁶.”)

[Reviewer:] In addition, the analysis of time lags in peak correlations, which forms the foundation for the authors existing interpretation, is not convincing for the following reasons:

- If stimulation was indeed causing head movements, this should be clearly apparent as changes in head movement parameters at a short latency following stimulation. But no plots of this kind are provided. The only plot provided is at a low time resolution, so the effect latency cannot be assessed.

Reply: It is not clear to us why the Reviewer states that no plots of this kind are provided. Figure 4c shows latencies between 40 - 80 ms. Figs. 4k,l show latencies in a similar range. We maintain that these latencies are what one would expect for a pathway connecting the sensory thalamus to head motor control. These latencies should be longer than those characterizing startle or other reflexive responses mediated via the brainstem (~ 30 ms; García-Hernández & Rubio, 2022).

In order to strengthen this point we have added four panels showing the effects of optogenetic stimulations on cross-correlations between single-units and head kinematics (Extended data figure 8). The four examples show peaks at the latency range, with

neurons both leading and lagging changes in linear velocity, and variable stimulation effects. A corresponding text was added (Lines 230-233) and legend of Extended data figure 8.

The time resolution is dictated by our video frame rate, which was 25 Hz in these experiments. This rate is above the Nyquist requirement for a system with a bandwidth of ~10 Hz.

[Reviewer:] First of all, there is no labeling of the time base in the inset of Figure 4c, so it is not clear where 40-80 ms is coming from. Second, according to the legend for 4c, it uses the data for the example presented in 4a. If it was clear where the 40-80 ms range was coming from, it would still not be apparent how reproducible this range is across replicates, nor the basis for the authors' statement that "neuronal changes typically led head velocity changes". Lastly, to accept that the causal lag here is meaningful, it would need to reflect a significant change from the correlation in the absence of stimulation, but that correlation is not shown.

REPLY: We thank the Reviewer for these helpful comments. First, we have replaced the inset of Figure 4c with a better version, including labels. Second, we have added "in this case" (Line 186) to clarify that our statements in this paragraph refer to the specific example. Third, we removed "by about 40 - 80 ms;" – the shape of the cross-correlation near zero is now clearly seen in the inset).

The last comment of the Reviewer is not clear to us. Causality in this example is demonstrated by the combination of panels a-c. We show the statistical significance of the occurrence of the causal lags (example of which is shown in panel c) in our experimental sample following stimulations later, in panels k-l, showing that the fraction of causal lags increased by our stimulations in a statistically significant manner.

[Reviewer:] The data in Figure 4k and l also do not unambiguously support the conclusion that stimulation specifically causes a change at 40-80 ms. The statistical test done to show a significant change between the control and stimulation distribution tests the entire distribution, not the values at 40 or 80 ms specifically – as far as I can tell from the Methods. Therefore, the differences could result, at least in part, from differences at other latencies. The distributions shown in 4k and l lack error bars, so the actual differences between conditions not due to chance are not clear. Perhaps the authors could do a test just on the numbers of cells that show peaks in a certain range or at a certain value, or treat each animal as a replicate and test the difference across animals. The examples added in ED Figure 8 help illustrate the measurements, but as examples, they do not address the significance of the changes.

REPLY: We are puzzled by this comment and suspect that the Reviewer might be misled by the phrase "by about 40 - 80 ms" attached to "neuronal changes typically led head velocity changes" cited above. As we explained above, these statements referred only to the example described in that paragraph (Fig. 4a-c). We now fixed this misleading text by

removing the phrase “by about 40 - 80 ms;” and adding “in this case” (Line 186) to clarify that our statements in this paragraph refer to the specific example.

Regarding Figures 4k and l. These two panels describe distributions. That is, for every value of Peak Offset the graph shows a single number which is the count of cases at this lag (more accurately, in the bin labeled by that lag). Each panel shows the distributions of lags for both the natural and stimulation conditions. No error bars can be plotted here because in each distribution there is only one value for every lag – the count. Similarly, no specific significance test can be performed for selected lags (for every lag there is only one data point in each graph). The only significant test possible is the comparison of the two entire distributions.

We thank the Reviewer for the specific suggestions but we can't see how they could work. If we select only those cells that show peak at the “right lag” during stimulations than the test is circular. Also we can't see how treating each animal as a replicate helps here. We indeed believe that the only valid test here is the comparison of the two distributions.

[Reviewer:] In general, the expected latency depends on the pathways mediating the effects. If the latency of the effect here cannot be distinguished from the latency of nonspecific or indirect effects, this challenges the interpretation that stimulation is revealing pathways that directly link whisker thalamus to head movements.

REPLY: As we said above, we do not claim here that our results prove such a direct link. We indeed believe that the answer to this question requires a new detailed study. Yet, our results do eliminate two major alternatives – those suggested by the two Reviewers. As we said in our reply in the previous round, the latencies we observe here (40-120 ms) are different from those expected in a startle response and from those expected from perceptual reports or perception-based decisions.

- Looking for signs of direct causal relationships using the shape of cross-correlation functions is rather indirect. The authors cite an effect latency with reference to the peak in such a plot, but the correlation values are not much lower at surrounding latencies. The correlation at 0 lag looks to be less than 10% different, and the values at nearby acausal lags, or causal lags that are long and non-specific, look to only be about 20% different. Moreover, the function is not directly compared to a control curve, and no statistics are provided.

Reply: The Reviewer probably misses the correct interpretation of the cross-correlation panel. The surrounding latencies that should be compared to the peak correlation are not those of the surrounding peaks, but rather those of the surrounding troughs. The surrounding peaks reflect mapping of the 2 Hz stimulation autocorrelation onto the crosscorrelation. The correct comparison shows a difference of about 50% (note that this is taken relative to the baseline of the 0.5 Hz periodic correlation).

We should remind the Reviewer that these data are recorded in actively moving, head-free unrestrained animals, in which signal-to-noise ratios for responses to external stimulations are much lower than those expected in passive modes, such as in head-fixed

conditions. Given that, we consider the signal-to-noise ratios shown here as high ratios. Finally, the only point of this panel is to show the latency, which, in our eyes, is presented clearly in the panel. Thus, to the best of our judgement, no shuffled correlation is required for convincing that the latency of the central peak was the mean latency between the firing rate and head linear speed in this experiment.

[Reviewer:] The authors have not understood my concern. I was not referring to the surrounding peaks, but to surrounding points. The inset in 4c shows that the curve is very nearly symmetric around zero. That the peak falls slightly to the right of 0 using the data from the example shown in panel a (this is according to the legend) is not statistically founded evidence that spiking in the thalamic cells leads head movement at a short causal lag. Replicating this measurement across animals (the authors refer to measuring the effects “in several cases”) would provide a statistical foundation for the statement about the 40-80 ms causal lag in the Results. This should not require collecting more data.

REPLY: We thank the Reviewer for the clarification. As explained above, we have replaced the inset of Figure 4c with a better version, which better represents the data. We also removed the confusing “by about 40 - 80 ms;” from that paragraph. Also, as we said above, we show the statistical significance of the occurrence of such causal lags in our experimental sample following stimulations later, in panels k-l, showing that the fraction of causal lags increased by our stimulations in a statistically significant manner.

- The authors also support their claim using reference to the fraction of cells whose spiking has a peak correlation with the head movement parameters at causal lags, citing increases from values near 30% to values near 50%. There are two problems here. First, no statistics are provided to indicate if these changes are significant. But much more importantly, under a null hypothesis of no causal relationship, the value should approach 50%, since spiking would be unrelated to behavior, and so correlation peaks should be on either side of 0 with equal likelihood. Because we are adding a lot of new spikes here via the stimulation, the relevant comparison is not necessarily the fraction obtained without stimulation. If the new, Chr2-induced spikes are dominating cell firing but induced spiking is unrelated to behavior, we expect 50% causal peaks. This could be addressed with proper statistics, but none are provided.

Reply: “First, no statistics are provided to indicate if these changes are significant.” – we do provide statistics (Legend of Figure 4k,l; Lines 486-487 of submitted version): “Free behavior and stimulation distributions were significantly different for VH ($p = 0.03$) and not for θ_H ($p = 0.36$); permutation tests, see Methods”

[Reviewer:] I thank the author for pointing this out – I had missed the statistical tests quoted in the legend. However, as I describe above, because of the test used, it does not specifically indicate a significant change in neurons with firing rates leading head movement.

REPLY: We believe that we have addressed the remaining concern of the Reviewer in our REPLY to their comment above (starting with “The data in Figure 4k and l also do not unambiguously support...”).

“under a null hypothesis of no causal relationship” – we do not agree with the Reviewer here. Our null hypothesis here is that there is a sensory causal relationship here and no motor relationship. We do not start from scratch here – we do know, based on the data presented up to this point in the paper, that thalamic neurons do respond to changes in head kinematics. What we do not know is whether they also affect head kinematics. Thus, the assertion of the Reviewer that “If the new, ChR2-induced spikes are dominating cell firing but induced spiking is unrelated to behavior, we expect 50% causal peaks.” Is wrong, because we do know that before adding new ChR2-induced spikes most of the thalamic spikes lag the changes in head kinematics.

[Reviewer:] Respectfully, my assertion quoted at the end there is not wrong. The spikes induced by ChR2 stimulation are not caused by sensation, so there are two possibilities for those spikes – they can cause movement, or they can be unrelated to movement. If the former, we expect a causal lag between spike times and head movement. If the latter, we expect the spikes to be equally likely at causal and acausal lags. Note that here by “acausal” I mean from the perspective of the neuron – you could replace these terms with “motor” and “sensory” if you prefer. Now, imagine a scenario in which during normal behavior, firing is driven by sensation, and so spiking lags head movements on average. If we induce spiking with ChR2 in very large amounts relative to the firing during behavior so that the vast majority of spikes are induced by ChR2, what do we expect under our two scenarios? If the induced spiking is driving head movements, we expect spiking mostly at a causal lag. If the induced spiking and head movements are not causally related, we expect an equal number of causal and acausal lags.

Therefore, in order for a change in the number of causal peaks from 30% to 50% to be evidence of causal influence, we need to know how much of the spikes in these calculations are induced, versus naturally occurring. The authors should be able to estimate this to provide evidence that the induced spikes are primarily at causal lags.

That the induced spikes do not dominate does not look to be easily assumed. The example shown in 4a shows that the firing rates are much higher during stimulation. The pulses are brief, which might mean that induced spikes do not meaningfully dominate here, but the authors would need to show that.

REPLY: We thank the Reviewer for the clarification. We did not interpret correctly the Reviewer’s stated premise saying” *If the new, ChR2-induced spikes are dominating cell firing...*”. We assume that the Reviewer means here “dominating by far”, such that the initial sensory-preferring distribution of lags is negligible. In that case we agree with the Reviewer’s rational and apologize for labeling their assertion as wrong.

To address this concern, we evaluated the fraction of spikes contributed to our thalamic neurons by our laser stimulations. To be conservative in our estimate, we assessed the number of spikes added by laser stimulations during our tagging procedures, which were conducted under anesthesia, a state characterized by low background neuronal activity (and thus over-weighting the contribution of stimulation-evoked spikes). During the tagging procedure, the number of spikes evoked by laser stimulations in each tagged neuron comprised, on average, 0.52 ± 0.29 (mean \pm std) of the spikes produced by that neuron during the stimulation period.

This is indeed an important analysis and we thank the Reviewer for suggesting it. This analysis shows that the evoked spikes did not over dominate neuronal activity, thus did not cancel the sensory bias observed before the stimulation. We have added this information in Lines 879-882.

- The authors refer to changes in correlation peak latency in general and refer to Extended Data Fig. 7, but this figure shows essentially no change for VPM and only shows changes for POm. Moreover, the correlation coefficients peak on either side of 0 for the two parameters shown in relation to POm spiking. Why would a stronger correlation be seen upon stim in the acausal direction if the effect was causal? This is not addressed.

Reply: Extended Data Fig. 7 was added in order to demonstrate the spectrum of correlation changes observed in the data. In order to address the concern of the Reviewer we have now added Extended Data Fig. 8, which shows several types of changes in correlations between individual neurons and head kinematics (linear velocity). The new figure shows an emergence of a correlation peak upon stimulation and strengthening of existing peaks, in both the motor and sensory causal directions. Please note that there is no causal and acausal directions here – both sensory and motor directions are causal within the motor-sensory loop.

[Reviewer:] The authors have not addressed the concerns that the correlations do not seem to change for VPM upon ChR2 stimulation (Extended Data Figure 7a,c) in the range (< 200 ms) stated in the text that references this figure (lines 215-218).

It is also not clear exactly what structure in Extended Data Figure 7b,d supports the authors claim either. Panel b does not seem to show an increase in correlation biased in the causal direction, and so seems equally consistent with an absence of causal effects on head movements. If the only support for the authors claim comes from panel d, the text should make that clear, and the relevant structure in panel d that supports that should also be clarified. Since the chance distributions for the Behavior and Stimulation data are so different (likely because much more Behavior data is used), it is hard to assess the real differences in the two curves plotted.

REPLY: We thank the Reviewer for this observation. Indeed our phrasing was not accurate enough. The Figure intended to show both changed and unchanged cases and

both sensory-like and motor-like changes, but our text failed to describe it correctly. This text now reads (Lines 214-218): “Thalamic neuronal populations exhibited correlations whose peak delays typically led or lagged head kinematic changes by < 200 ms (**Extended data figure 7**). In some cases, neuronal-head kinematic correlations could be modified by optogenetic stimulations, adding either sensory-like or motor-like neuronal activity (**Extended data figure 7b and d**, respectively).”

The support to our claim does not come from panel d – panel d is an example. We use **Extended data figure 7** to show the variety of cases, the distributions of which are depicted in panels k and l. The support to our claim comes from the statistical significance of the difference in peak delays, presented in panels k and l.

- The language the authors have added to lines 219-225 to clarify their interpretation does not seem sufficiently clear. Are they referring to the causal latencies, or the acausal latencies, that could also reflect expected sensory responses? The inclusion of “and perceiving” here in a section that seems oriented toward motor responses seems to confuse matters.

Reply: this is now changed to: “It is possible that the observed shift in relative timing between neuronal activity changes and head kinematic changes reflect two unrelated processes, both induced by the stimulations. Yet, it is important to note that the delays observed here between neuronal and kinematic changes (typically between 40 and 120 ms) are longer than those characterizing brainstem-based startle responses³⁴ and shorter than those characterizing cortex-based perceptual reports³⁵. These delays are consistent with the scheme in which the VPM and POM, considered thus far to focus on the processing of sensory vibrissal information, are active players in the sensory-motor loop, controlling and perceiving head kinematics.” (Lines 228-235)

[Reviewer:] As I describe above, distinguishing from the latencies of naturally occurring startle is not sufficient to relieve my concern. I would also note that natural startles would involve sensory reception and peripheral conduction, which would not be part of the latency here. The larger problem here though is that a statistical foundation for the observation of increased causal correlation is not sufficiently strong. I do believe it may be possible to present a sound statistical argument with the existing data however.

REPLY: We hope that the additional analysis we performed (addressing the comment starting with “Respectfully, my assertion quoted...” above) satisfies the Reviewer’s requirement of a stronger statistical argument.

- If we want to make an argument that these thalamic regions are driving head movements, it would also seem logical to look for evidence in the relationship between naturally occurring firing and head (or whisker) movement parameters. However, no such analysis is presented.

Reply: The Reviewer is not correct here. Fig. 4h-l panels all include “behavior” and “stimulation” data. “Behavior” refers to data during natural behavior.

[Reviewer:] My general point is that it would be helpful to show that naturally occurring activity is consistent with causal influence on head movement, and not just implicitly. The author is technically correct in their reply here, but these data are all presented in the context of the optogenetic results. 4h is at low time resolution, so the Behavior curve is hard to read at the relevant lags. 4i is unrelated to neural activity. 4j combines data from both free behavior and stimulation, such that it is not clear that causal lags are present during free behavior. The one panel that does include cells showing causal lags during free behavior is described only with the observation that most neurons showed acausal/sensory lags, as a counterpoint to the results in the next panel.

REPLY: We have replaced Fig.4h with a version with a higher time resolution, showing lags between -1 and 1 s. Regarding the other panels, it's true that each one presents a different aspect of the data. We believe that it is essential for our paper to display all these perspectives, including kinematic correlations, neuronal-to-kinematic correlations for each kinematic variable, and correlations during both free behavior and stimulation. The panel that shows neuronal-kinematic correlations during free behavior (panel k) indeed shows that most of the neurons exhibit sensory (“acausal” in motor terms) lags – this is what we state in the text (Lines 223-4 “During free exploration, activity changes in most neurons lagged changes in $\dot{\theta}_H$ and in V_H (**Fig. 4k**).”). The next panel indeed indicates a shift in this pattern during stimulation, with more neurons displaying motor-causal lags (“The relative fraction of neurons that lead changes in $\dot{\theta}_H$ and V_H significantly increased with optogenetic stimulations (**Fig. 4l**)...”). We must acknowledge that we do not fully understand the Reviewer’s concern here, as we believe our statements are well-supported by the data presented in these panels.

Reviewer #3 (Remarks to the Author):

The revised manuscript of Oram, Tenzer et al. is modestly improved but the central issues identified by myself and the other reviewers largely remain. The authors’ response mostly presents arguments with critiques of the reviewers in lieu of improvements to the manuscript, and in most cases, I do not find these arguments compelling. The central result regarding coding of head kinematics still suffers from the issue of correlations between kinematic variables, and so it is not clear that the authors have correctly identified that head kinematics are the thing being encoded. Demonstration of allocentric coding of whiskers remains unconvincing. Some results of the study may well be correct, but it is impossible to know with any certainty given the data and analyses presented.

Finally, all three reviewers noted that the results of Figure 4 should not be interpreted in the way the manuscript interprets them. After two rounds of reviews, it remains essentially unchanged. All three reviewers articulated these concerns – albeit in different

ways – but all comments were fundamentally focused on the same problem. I very much agree with the statements made by the other reviewers and echoed my own. Because the problem was described in different ways by the different reviewers, the Authors make it seem like different criticisms were levied by different reviewers on different rounds of review, but I do not see it this way. In the latest revision, the authors interpret concerns of myself and R2 overly narrowly so as to refute them, for example, with respect to addressing the startle-like responses mentioned by R2. R2’s point persists if the response to optogenetic stimulation is not a literal startle response mediated only by medullary circuits, but merely startling to the animal in some way. The authors seem to suggest, in response to my previous review, that the response of a mouse to external stimuli can only fall into the categories of ‘startle responses’ and ‘perceptual decisions’ with latencies >150 ms. Mice can certainly respond to external stimuli with learned movements in 50-100 ms and innate movements to strong stimuli are likely faster. A mouse orienting to identify what just touched its tail certainly does not require a perceptual decision in the sense that that term is usually used in the literature. I don’t know why ‘significant cortical processing’ would be assumed.

REPLY: We thank the Reviewer for their clear and constructive summary of their concerns. We acknowledge the ongoing debate regarding Figure 4 and the interpretation of our optogenetic stimulation results. Consequently, we have decided to revise our interpretations and significantly moderate our stance on the motor-control interpretation.

In fact, thanks to the Reviewers’ (justified) concerns, we realized that our use of the terminology of “motor control” is not justified and indeed might be misleading. Thus, throughout the paper, we have replaced this terminology with one expressing the potential effect of thalamic activity on head kinematics, better conveying the proposed function of the sensory thalamic nuclei in active perception of head kinematics. We also emphasize now that more definitive answers require further experimentation. The actual changes are (the essential changes are underlined):

1. End of Abstract (Lines 21-24): we changed “**Collectively, our data suggest that the VPM and POM both process ... and participate in the control of head motion.**” to “**Collectively, our data suggest that the VPM and POM both process ... and can affect the dynamics of head kinematics.**” (the addition of “can” here emphasizes that the animals do not necessarily use this ability during natural behavior).
2. Lines 172: we changed “In order to study the involvement of these thalamic neurons in head-motion control loops, loops in which these neurons both process sensory information and affect motor activity, we tracked ...” to “In order to test whether these thalamic neurons can affect head kinematics, we tracked ...”
3. Lines 222-223: we changed “This suggests a scheme in which thalamic neurons of the vibrissal sensory nuclei participate in functional loops that control head kinematics” to “This suggests a scheme in which thalamic neurons of the vibrissal sensory nuclei participate in functional loops that affect head kinematics”.
4. Lines 233-236: changed “These delays are consistent with the scheme in which the VPM and POM, considered thus far to focus on the processing of sensory

- vibrissal information, are active players in the sensory-motor loop, controlling and perceiving head kinematics.” To “These delays are consistent with the possibility that the VPM and POm, considered thus far to focus on the processing of sensory vibrissal information, might be active players in the sensory-motor loop, mediating the perception of head kinematics.”
5. Line 245: deleted “Not only that, but the activity profile of VPM and POm neurons is consistent with these nuclei participating in sensory-motor loops controlling head motion (the reasonable possibility that they also participate in loops controlling whisking was not tested here).”
 6. Line 284-287: changed “further investigations are warranted to unravel the specific dynamical processes involved in object perception within these nuclei and to elucidate the division of labor between them.” to “further investigations are warranted to unravel the specific dynamical processes involved in object perception within these nuclei, to elucidate the division of labor between them and to clarify their potential effects on head and whiskers kinematics.”

In addition, we would like to address here the specific concerns expressed by the Reviewer.

“R2’s point persists if the response to optogenetic stimulation is not a literal startle response mediated only by medullary circuits, but merely startling to the animal in some way” – we respectfully do not agree that R2’s point still persists in this case. We believe that ruling out literal startle response and perceptual reports suggests a dynamical process that differs from these two, a process that may facilitate perceptual convergence. Such a process may indeed resemble a startling response in some way, but such resemblance would not classify it as a non-perceptual process.

“Mice can certainly respond to external stimuli with learned movements in 50-100 ms and innate movements to strong stimuli are likely faster” – we are not sure to what literature the Reviewer refers here. We are not familiar with any study showing 50-100 ms responses involving head movements in mice. As for innate perceptual responses (e.g. to visual stimuli), the literature we are familiar with demonstrate delays > 200 ms (e.g., Yilmaz & Meister, Curr Biol 2013, Rapid Innate Defensive Responses of Mice to Looming Visual Stimuli). Regarding learned responses – it is important to note that no learning was induced in our protocols.

“A mouse orienting to identify what just touched its tail certainly does not require a perceptual decision in the sense that that term is usually used in the literature.” – We agree of course. As stated above, we believe that ruling out startle responses and perceptual decisions suggests a dynamical process that differs from these two, a process that may facilitate tactile perceptual convergence.

REVIEWER COMMENTS

Reviewer #2 (Remarks to the Author):

The revision of language in the text has improved the manuscript and has to some extent addressed my concerns about its interpretations of data. I have the following remaining concerns:

Line 186: “neuronal changes typically led head velocity changes (Fig. 4c)” - This statement does not seem well supported by the data provided, given that the plot in 4c shows only an individual example, and in that example, the distribution straddles 0, with a peak only slightly to the right of zero. “Typically” would seem to imply that the observation is repeated, but no supporting statistics are provided. I understand some statistics are done on the data shown in Figure 4k and I, but (a) that is not raised for another 45 lines of text, and (b) the statistics there support a different statement, not that “neuronal changes typically led head velocity changes.”

Line 224: As far as I can tell from the figure legend and methods, the significance test used to compare the distributions in Figure 4k and I was comparing the entire distributions, which span from -600 to 600 ms, including both leading and lagging latencies. Yet the authors state that “The relative fraction of neurons that lead changes in θ_H and V_H significantly increased with optogenetic stimulations.” I do not see the justification for saying that the number that lead specifically increases. This should be easily addressed by testing the distributions specifically on one side of 0.

To aid the reader in interpreting the increases in the fraction of neurons whose firing leads θ_H and V_H , the authors should also report the increase in firing rate between the behavior and stimulation epochs used for the analysis. The measurements added to lines 879-882 are good, but are not what would be most directly relevant. These firing rate values are needed in the main text so the reader can discern what fractional increase would be expected under the null hypothesis that opto-induced spikes are unrelated to head movement (equally leading and lagging).

Line 233: “might be active players in the sensory-motor loop, mediating the perception of head kinematics.” This phrasing is not clear. The section began with the stated goal to “test whether these thalamic neurons can affect head kinematics” and so the relevant question is whether there is influence on head movement, not just whether the thalamic regions in question merely mediate the perception of head kinematics. The focus on causal lags in this section suggests a motor involvement, but “perception” makes it sound like the authors are referring to a sensory role.

Reviewer #3 (Remarks to the Author):

The revised manuscript of Oram et al. again contains minor improvements over previous versions. Many of the concerns raised in previous rounds of reviews still remain unaddressed, however. A brief summary of a few of the major issues that continue to remain are below. Please see my previous several critiques for more details on each of them.

- 1) Shuffled distributions still appear to be incorrectly computed.
- 2) Correlations between kinematic variables have not been meaningfully addressed.
- 3) Allocentric tuning is not treated with the level of detail needed to make meaningful statements.
- 4) Figure 4 remains problematic. I acknowledge that the authors did add qualifiers to their description of these results, but I don't understand how conclusions such as that stimulation of VPM, POM "can affect the dynamics of head kinematics" are important contributions to the literature. If you activated cutaneous receptors on the tail, it could affect head kinematics. If one stopped the animal's heart from beating, it could affect head kinematics. But these effects would be trivial. I don't see how these results can be interpreted in any meaningful way.

Rebuttal point-by-point reply to reviewers' comments

The comments of the Reviewers appear in black. Our replies are in blue text and start with "REPLY:"

Reviewer #2 (Remarks to the Author):

The revision of language in the text has improved the manuscript and has to some extent addressed my concerns about its interpretations of data. I have the following remaining concerns:

Line 186: "neuronal changes typically led head velocity changes (Fig. 4c)" - This statement does not seem well supported by the data provided, given that the plot in 4c shows only an individual example, and in that example, the distribution straddles 0, with a peak only slightly to the right of zero. "Typically" would seem to imply that the observation is repeated, but no supporting statistics are provided. I understand some statistics are done on the data shown in Figure 4k and l, but (a) that is not raised for another 45 lines of text, and (b) the statistics there support a different statement, not that "neuronal changes typically led head velocity changes."

REPLY: We agree. First, the term "typically" was ambiguous in this sentence. Here it meant that neuronal changes tended to lead head velocity changes IN THIS EXAMPLE (not among all studied cases – this is indeed treated later in the paper). To emphasize the reference to this example we have added "in this case" just before (Line 186) but this indeed might be read as referring only to the statement of intensity correlations. This is now fixed in the new phrasing.

Second, indeed statistical data supporting the leading tendency in this example was missing. We have thus added a permutation test to support this statement statistically. For symmetry, we also added a statistical test to support the correlation presented in Fig. 4b. The modifying sentence now reads (Lines 184-187): "As expected from a process in which changes in thalamic firing take part in changing head velocity, in this example the intensities of changes were correlated (**Fig. 4b**; $r^2 = 0.2$, $p = 0.001$) and neuronal changes tended to lead head velocity changes (**Fig. 4c**; $p = 0.009$, permutation test)."

Regarding the Reviewer's concern implied in the statement "the distribution straddles 0, with a peak only slightly to the right of zero" we would like to state that, based on our experience, a cross-correlation function that straddles 0 is what we expect when cross-correlating two dynamical stochastic processes that are intermingled with other related dynamical stochastic processes. And the delay of the peak (order of 120 ms) is the delay expected from several pathways that are candidate pathways for closing this sensory-motor loop.

Line 224: As far as I can tell from the figure legend and methods, the significance test used to compare the distributions in Figure 4k and l was comparing the entire distributions, which span from -600 to 600 ms, including both leading and lagging latencies. Yet the authors state that "The relative fraction of neurons that lead changes in θ_H and V_H significantly increased with optogenetic stimulations." I do not see the

justification for saying that the number that lead specifically increases. This should be easily addressed by testing the distributions specifically on one side of 0.

REPLY: We agree. We thus performed the analysis suggested by the Reviewer and corrected the text in the Figure legend accordingly. It now reads (Lines 526-527): “The increase in prevalence of positive peak delays (between 0 and 400 ms) was statistically significant for V_H ($p = 0.02$) and not for θ_H ($p = 0.16$) (Permutation tests, see Methods).

To aid the reader in interpreting the increases in the fraction of neurons whose firing leads θ_H and V_H , the authors should also report the increase in firing rate between the behavior and stimulation epochs used for the analysis. The measurements added to lines 879-882 are good, but are not what would be most directly relevant. These firing rate values are needed in the main text so the reader can discern what fractional increase would be expected under the null hypothesis that opto-induced spikes are unrelated to head movement (equally leading and lagging).

REPLY: We agree. We have added the following text in Lines 228-235: “It is important to mention that, while being effective, the spikes induced by our optogenetic stimulations did not over dominate neuronal activity, and thus likely did not cancel the sensory bias observed before the stimulation. During anesthesia, the number of spikes evoked by laser stimulations in each tagged unit comprised, on average, 0.52 ± 0.29 (mean \pm std) of the spikes produced by that neuron during the stimulation period (see Methods). During behavior, laser stimulations increased the mean firing rate of tagged units, on average for the entire stimulation block, by 15.5 ± 14.5 % (mean \pm std).”

Line 233: “might be active players in the sensory-motor loop, mediating the perception of head kinematics.” This phrasing is not clear. The section began with the stated goal to “test whether these thalamic neurons can affect head kinematics” and so the relevant question is whether there is influence on head movement, not just whether the thalamic regions in question merely mediate the perception of head kinematics. The focus on causal lags in this section suggests a motor involvement, but “perception” makes it sound like the authors are referring to a sensory role.

REPLY: We agree. This sentence was modified to read (Line 240-242): “In contrast, these delays are consistent with the possibility that the VPM and POm, considered thus far to focus on the processing of sensory vibrissal information, also play an active role in sensory-motor loops of head kinematics.”

Reviewer #3 (Remarks to the Author):

The revised manuscript of Oram et al. again contains minor improvements over previous versions. Many of the concerns raised in previous rounds of reviews still remain unaddressed, however. A brief summary of a few of the major issues that continue to remain are below. Please see my previous several critiques for more details on each of

them.

1) Shuffled distributions still appear to be incorrectly computed.

REPLY: In the first review round, we addressed the Reviewer's concern about shuffled data in Figures 2b,c and 3c,e, that "in the shuffled tuning curves, the neuron's firing rate ought to be a little above the minimum firing rate at all azimuthal velocities – not near the maximum firing rate". We explained our method (normalization to max), and added explanatory text to the relevant legends. The added text was apparently not clear enough, and we have now modified it to read (legends of Figs. 2b, 3c,e): "Each curve, of both empirical and shuffled data, is normalized to its maximal value; the firing rates labeling the ordinate in each plot refer to the empirical data only." We believe that this text now clarifies what is presented in the figure panels and precludes ambiguities.

2) Correlations between kinematic variables have not been meaningfully addressed.

REPLY: In the previous round, the Reviewer said: "The central result regarding coding of head kinematics still suffers from the issue of correlations between kinematic variables, and so it is not clear that the authors have correctly identified that head kinematics are the thing being encoded." We addressed this comment in length in the review iteration before last, and never received a direct objection to our reply. We accept that there is a disagreement between us and the Reviewer on this issue, but do not know how to reconcile it, or add anything new without understanding exactly what is wrong with our reply.

3) Allocentric tuning is not treated with the level of detail needed to make meaningful statements.

REPLY: We fully agree with the Reviewer that allocentric tuning deserves a deep and serious treatment. But we do not agree with the requirement that such a deep treatment should be done in the framework of the current paper. In the first review round the Reviewer suggested that: "Minimally, tuning curves need to be included." We accepted this requirement and added 16 tuning curves for allocentric variables - 8 per nucleus and 4 per variable (Extended data figure 9). Addressing the minimal requirement of the Reviewer we believe that this can be considered sufficient for the current paper, leaving a deeper treatment to a follow up study.

4) Figure 4 remains problematic. I acknowledge that the authors did add qualifiers to their description of these results, but I don't understand how conclusions such as that stimulation of VPM, POm "can affect the dynamics of head kinematics" are important contributions to the literature. If you activated cutaneous receptors on the tail, it could affect head kinematics. If one stopped the animal's heart from beating, it could affect head kinematics. But these effects would be trivial. I don't see how these results can be interpreted in any meaningful way.

REPLY: We agree that the statement is not specific enough. We thus changed our statement to more specifically relate to the tight whisker and head coordination during active perception. The statement now reads (Lines 21-24): “Collectively, our data suggest that the VPM and POm both process combined head and whisker information – an integration that is consistent with the tight head and whisker coordination during active perception – and can affect the dynamics of head kinematics during tactile perception.” The main point here is the following: While activation of tail cutaneous receptors may affect head kinematics, this would be an ‘open-loop’ effect. There will be no direct effect back from head movements to tail tactile receptors. The case of POm and VPM activations is entirely different –their activations affect head movements, thereby closing a loop, in which VPM and POm neurons will be affected by head movements. Our paper does not dwell on these closed-loop dynamics, thus we did not find it appropriate to expand on this issue in the Abstract. We thank the Reviewer for this important comment and hope that the additional specific addition (“during tactile perception”) addresses their concern.